



# Characterising the surface microlayer in the Mediterranean Sea: trace metals concentration and microbial plankton abundance

Antonio Tovar-Sánchez[1], Araceli Rodríguez-Romero[1], Anja Engel[2], Birthe Zäncker[2], Franck Fu[3], Emilio Marañón[4], María Pérez-Lorenzo[4], Matthieu Bressac[5,6], Thibaut Wagener[7], Karine Desboeuf[3], Sylvain Triquet[3], Guillaume Siour[3], Cécile Guieu[7]

[1]Department of Ecology and Coastal Management, Institute of Marine Sciences of Andalusia (ICMAN-CSIC), Puerto Real, 07190, Spain
[2]Leibniz Institute of Marine Sciences (IFM-GEOMAR), Kiel, Germany
[3]Laboratoire Interuniversitaire des Systèmes Atmosphériques (LISA), CNRS UMR 7583, Université de Paris, Université Paris-Est-Créteil, Institut Pierre Simon Laplace (IPSL), Créteil, 94000, France
[4]Departamento de Ecología y Biología Animal, Universidad de Vigo, 36310 Vigo, Spain
[5]Institute for Marine and Antarctic Studies, University of Tasmania, Hobart, Tasmania, Australia
[6]Sorbonne Université, CNRS, Laboratoire d'Océanographie de Villefranche, LOV, F-06230, Villefranche-sur-mer, France
[7]Aix Marseille Univ., CNRS, IRD, Université de Toulon, MIO UM 110, 13288, Marseille, France

*Correspondence to*: Antonio Tovar-Sánchez (a.tovar@csic.es)

## Abstract

The Sea Surface Microlayer (SML) is known to be enriched in trace metals relative to the underlaying water and to harbor diverse microbial communities (i.e. neuston). However, the processes linking metals and biota in the SML are not yet fully understood. In this study, we analyzed the metal (Cd, Co, Cu, Fe, Ni, Mo, V, Zn and Pb) concentrations in aerosol samples, SML (dissolved and total fractions) and in subsurface waters (SSW; dissolved fraction at ~1 m depth) from the Western Mediterranean Sea during a cruise in May-June 2017. The bacterial community composition and abundance in the SML and SSW, and the primary production and Chl-a in the SSW were measured simultaneously at all stations during the cruise. Residence times of particulate metals derived from aerosols deposition ranged from a couple of minutes for Co ($2.7 \pm 0.9$ min; more affected by wind conditions) to a few hours for Cu ($3.0 \pm 1.9$ h). Concentration of most dissolved metals in both, the SML and SSW, were well correlated with the salinity gradient and showed the characteristic eastward increase in surface waters of the Mediterranean Sea (MS). Contrarily, the total fraction of some reactive metals in the SML (i.e. Cu, Fe, Pb and Zn) showed negative trends with salinity, these trends of concentrations seem to be associate to microbial uptake. Our results suggest a toxic effect of Ni on neuston and microbiology community's abundance of the top meter of the surface waters of the Western Mediterranean Sea.



## 1. Introduction

The Mediterranean Sea (MS) is enriched in many trace metals relative to similar nutrient- depleted waters in the open ocean (e.g. Cd, Cr, Co, Cu, Ni, Fe, Zn) (Bonnet et al., 2013; Boyle et al., 1985; Sarthou and Jeandel, 2001; Sherrell and Boyle, 1988). The enrichment of metals in surface water has been associated to different sources including atmospheric deposition, river inflows, groundwaters, anthropogenic sources and the Atlantic Ocean inflow through the Gibraltar Strait (Boyle et al., 1985; Elbaz-Poulichet et al., 2001; Migon, 2005; Trezzi et al., 2016). The MS has one of the highest rates of aeolian deposition in the world with strong pulses of mineral dust from Africa, in addition to consistent anthropogenic aerosol inputs from Europe. Therefore, atmospheric deposition, dry and wet, is the dominant pathway for large scale transport of trace metals to the water column and sediments in MS (Guieu et al., 2002, 2010; Jordi et al., 2012; Ternon et al., 2010; Tovar-Sánchez et al., 2010, 2014). Many of these metals play an important role in biogeochemical processes of this sea. For example, it has been hypothesized that the high Co concentrations in the MS stimulate "de novo" synthesis of vitamin $B_{12}$ as Co is the central metal ion in the $B_{12}$ molecule (Bonnet et al., 2013). Although present in higher concentration than in other oceans, Fe has been considered as an important factor controlling phytoplankton growth (Sarthou and Jeandel, 2001). Copper from aerosol depositions has been demonstrated to have toxic effects on marine phytoplankton (Jordi et al., 2012; Paytan et al., 2009) while Ni and Zn have been considered as good geochemical tracers of aerosols impact in *Posidonia oceanica* (Tovar-Sánchez et al., 2010).

Studying the Sea-Surface Microlayer (SML), especially in a region dominated by aeolian deposition, is crucial for understanding trace metal dust solubility, ocean distribution, and the processes influencing the primary production and the vertical particle fluxes in the water column. The SML is considered the skin of the ocean as it serves as a boundary layer between the atmosphere and the ocean. With a thickness of 1−1000 μm, it is a prevalent feature of the surface ocean that shows distinct physical, chemical, and biological properties than the rest of the water column. This sea-air interface plays a key role in regulating the exchanges of gases, solutes and energy between water and atmosphere and is central to a wide range of global biogeochemical and climate regulation processes (Cunliffe et al., 2013).



Characterized by the dominated abundance of microorganisms (called neuston and ranging from bacteria to larger siphonophores (Wurl et al., 2017)), the SML is a particular marine ecosystem ecologically regulated. Although neuston in SML is formed by recruitment from the underlying plankton, its composition and activities are different and conditioned by meteorological conditions, intensity of UV radiation, organic matter content and/or aerosol impact among others (Cunliffe et al., 2013; Engel et al., 2017).

Impacted by different allochthonous sources (e.g. aerosols, ice, rivers) the SML is enriched in reactive trace metals (e.g. Cu, Fe, Pb) which metal stoichiometry signature is different from underlaying waters (Tovar-Sánchez et al., 2019). For example, in regions under the influences of dust events such as the North Atlantic Ocean or Mediterranean Sea, concentrations of Cu, Fe or Pb in the SML increase by a factor of up to 800, 200 and 150 times compared with the underlaying water (Tovar-Sánchez et al., 2019). It has been estimated that the SML in the MS contents around 2 tons of total Fe, and this amount could be much higher during episodes of dust event (Tovar-Sánchez et al., 2014). However, despite of such enrichment in trace metals concentrations, little is known about their residence times, their influence on the active microbial community within the SML, or their release rates towards the underlaying waters. Previous studies, from field sampling and laboratory microcosms, have estimated the residence times of dissolved and particulate trace metals (e.g. Al, Fe, Mn, Ni, Cu, Zn, and Pb) in the SML to range from a few minutes to a few hours. This is likely long enough to be chemically and biologically alter the SML and affect the composition and activity of the neuston community (Ebling and Landing, 2017; Hardy et al., 1985). However, estimates of residence times considering a variety of key field measurements that directly affect the physical, chemical and biological composition of the surface microlayer, such as dry and wet deposition fluxes, wind speeds, and neuston composition, have not been addressed yet.

Here, we studied the dissolved (<0.22 µm) and particulate trace metal composition (Cd, Co, Cu, Fe, Ni, Mo, V, Zn and Pb) of the SML in the central and Western MS. Aerosols were sampled and analyzed for trace metals at the same stations and residence time of particulate aerosol metals in the SML calculated. We analyzed the microbial composition and abundance in the SML and subsurface water (SSW), and the primary production and Chl-a concentration in the subsurface water (1-5m), and examined the relationships with trace metals concentration and distribution.



## 2. Material and Methods

Samples from SML, SSW and aerosols were collected during the cruise PEACETIME (ProcEss studies at the Air-sEa Interface after dust deposition in the MEditerranean sea) on board the French R/V 'Pourquoi Pas?' in the MS, from May 10th to June 11th, 2017. Twelve stations were sampled (Figure 1). Three of these stations were sampled twice (TYR 1-2, ION 1-2) or five times (FAST 1-5) at different days, counting for a total of 17 groups of samples (Table 1).

### 2.1. Aerosol sampling and analysis

The PEGASUS container was installed aboard the R/V Pourquoi Pas?, this container is a mobile platform equipped with a set of instruments optimized to collect and analyze in real time, gaseous compounds and particles in the atmospheric boundary layer (Formenti et al., 2019). Atmospheric sampling was performed using isokinetic and wind-oriented aerosol multi-samplers with a total sampled flow rate around 400 L min$^{-1}$ per inlet. This inlet was developed for sampling both fine and coarse particles, with particles of aerodynamic diameter of about 40 µm (Rajot et al., 2008). This total flow was subdivided to various transmission lines which served the majority of the instrumentation. The aerosol size distribution from 10 nm to 30 µm was measured by a combination of standard optical and electrical mobility analyzers. The total mass concentration was obtained by an on-line Tapering Element Oscillating Microbalance (TEOM, model 1400a, Rupprecht and Patashnick).

One of the sampling lines was equipped with filtration unit to collect the aerosols on 47-mm polycarbonate membranes of 0.4 µm pore size (Whatman Nuclepore ™). The volume flow rate was set at 20 L.min$^{-1}$. All the filters were previously cleaned by immersion in ultrapure HCl (2%) during 2 hours and rinsing with ultrapure waters. A sampling strategy was made to avoid the contamination by the cruise smoking, firstly when the vessel was in station, the R/V was systematically positioned such as inlets were facing the wind (PEGASUS container and boat's chimney are on the opposite side of the deck). On the route, contamination-free sampling was operated when the relative wind direction was not in the direction of chimney exhaust. In total, 36 series of filters were collected which 17 filters during the stations and 5 blanks of filters were also prepared. The sampling locations for each filter is presented in Figure 1.





Aerosols filters were first analyzed by X-ray fluorescence spectrometry (SFX, spectrometer PW-2404, Panalytical™) for measuring chemical markers of particles origin sources (as Al and Ca). Filters were then leached by ultrapure water in order to determine the soluble fraction of metals. Finally, the filters were mineralized using an acid digestion protocol adapted from (Heimburger et al., 2013) in order to quantify the insoluble (particulate) fraction of metals. The dissolved and digested samples were analyzed by HR-ICP-MS (Neptune Plus, Thermo Scientific ™) for trace metals: Cd, Co, Cr, Cu, Fe, Mn, Mo, Nd, Ni, Pb, V, Zn. The total concentration of metals corresponds to the sum of dissolved and particulate fraction of metals. Rain sampling was also operated during the cruise with on-line filtration collector (0.2µm, polycarbonate, Nuclepore Whatman™) (Heimburger et al., 2013) and the analysis of particulate and dissolved metals were carried out by HR-ICP-MS.

## 2.2. Water sampling and analysis

### 2.2.1. Trace metals

Surface samples, i.e. SML and subsurface water (SSW: ~1 m depth), were collected from a pneumatic boat deployed 0.5 – 1 mile away from the research vessel in order to avoid contamination of the samples from the vessel's influence. SML samples were collected using a glass plate sampler (Stortini et al., 2012; Tovar-Sánchez et al., 2019) which had been previously cleaned with acid overnight and rinsed thoroughly with ultrapure water (MQ-water). The 39 x 25 cm silicate glass plate had an effective sampling surface area of 1950 cm$^2$ considering both sides. In order to check for procedural contamination, we collected SML blanks in some stations on board of the pneumatic boat by rinsing the glass plate with ultra-pure water and collecting 0.5 L using the glass plate system. The surface microlayer thickness was calculated following the formula of Wurl (2009) (Wurl, 2009). Total fraction of SML (i.e. T-SML) were directly collected from the glass plate system without filtration in a 0.5 L acid cleaned LDPE bottles, while that the dissolved fraction in the SML (i.e. D-SML) was rapidly filtered on board the pneumatic boat through an acid-cleaned polypropylene cartridge filter (0.22µm; MSI, Calyx®). SSW were collected using an acid-washed Teflon tubing connected to a peristaltic pump and directly filtered on the same cartridge to collect the dissolved fraction (D-SSW). All samples were acidified on board to pH< 2 with Ultrapure-grade HCl in a class-100 HEPA laminar flow hood. Metals (i.e. Cd, Co, Cu, Fe, Ni, Mo, V, Zn





and Pb) were pre-concentrated using an organic extraction method (Bruland et al., 1979) and quantified by ICP-MS (Perkin Elmer ELAN DRC-e). Prior to the preconcentration and for the breakdown of metal-organic complexes and the removal of organic matter (Achterberg et al., 2001; Milne et al., 2010), total fraction samples (i.e. T-SML) were digested using an UV system consisting in one UV (80 W) mercury lamp that irradiated the samples (contained in quartz bottles) during 30 min. The accuracy of the pre-concentration method and analysis for trace metals was established using Seawater Reference Material for Trace Elements (CASS 6, NRC-CNRC) with recoveries ranging from 89% for Mo to 108% for Pb.

### 2.2.2. Ancillary parameters

Temperature and Salinity of surface seawater were measured with the underway thermosalinograph (TSG) system of the R/V Pourquoi Pas? which was composed of a *Seabird® SBE 21 seaCAT* associated to a *SBE 38* thermometer situated at the seawater inlet. The seawater inlet was situated 3 meters under the sea surface. The wind speed at 10 meters was measured with a *Gill Windsonic* ultrasonic anemometer from the on-board BATOS station deployed by the French meteorological agency *Météo France* on the vessel. Temperature, salinity and wind data were averaged on 30 seconds timelapses by the ship data management system TECHSAS (*TECHnical Sensor Acquisition System*). Average values of temperature, salinity and wind speed on a time period of 1 hour around the time of SML sampling are reported in table 1.

### 2.3. Biological sampling and analysis

2.3.1. Neuston

Microorganism inhabiting the SML are collectively referred to as the "neuston" (Engel et al., 2017). At the same time than trace metal samples collection, microorganism in the SML were sampled also using a glass plate system (50 x 26 cm silicate glass plate with an effective sampling surface area of 2600 cm$^2$ considering both sides). The water from the SSW was manually collected in acid clean borosilicate bottles at around 20 cm depth. Bacterial numbers were determined using flow cytometry from a 4 mL sample that was fixed with 200 mL glutaraldehyde (GDA, 1% final concentration). Samples were stored at -20 ºC for at most 2.5 months until analysis and were stained with SYBR Green I (Molecular



Probes) prior to quantification using a flow cytometer equipped with a 488 nm laser (Becton & Dickinson FACScalibur). A plot of side scatter (SSC) vs. green fluorescence (FL1) was used to detect the unique signature of the bacterial cells. The internal standard consisted of yellow-green latex beads (Polysciences, 0.5 mm). Abundance and area of Transparent Exopolymer Particles (TEP) were measured

microscopically following a previously described method (Engel, A., 2009).

### 2.3.2. Phytoplankton and Primary Production

Chl-a concentration and primary production were measured in the SSW at 5 m depth. Primary production was measured with the $^{14}$C-uptake technique. Seawater samples, collected from Niskin bottles

at dawn, were dispensed into four (3 light and 1 dark) polystyrene bottles of 70 mL in volume, which were amended with 15 µCi of NaH$^{14}$CO$_3$ and incubated for 24 h inside a deck incubator refrigerated with surface seawater from the continuous water supply. The incubator was covered with a neutral density filter that provided an irradiance level of 70% of incident PAR. After incubation, samples were filtered, using low vacuum pressure, through 0.2-µm polycarbonate filters, which were exposed to HCl fumes

overnight to remove non-fixed, inorganic $^{14}$C. After adding 5 mL of liquid scintillation cocktail to the filters, the radioactivity on each sample was determined on-board with a liquid scintillation counter. To compute the rates of carbon fixation, the dark-bottle DPM value was subtracted from the light-bottle DPM value and a value of 26,000 µgC L$^{-1}$ was used for the concentration of dissolved inorganic carbon. Chla concentratios were measured by HPLC (HPLC Agilent Technologies 1200) following the method

described by Ras et al. (2008) (Ras et al., 2008).

### 2.4. Statistical analyses

Spearman rank correlation coefficient ($r_s$) was used to determine significant relationships ($p <$ 0.05) between the parameters measured in the different compartments (air, SML and SSW) and

parameters. Coefficient of determination ($R^2$) between the selected parameters were also calculated in order to determine how well correlations fit with a linear regression relationship. Statistical analysis was performed with the aid of the statistical software package SPSS 25.



## 3. Results and Discussion

### 3.1. Aerosols deposition

Metal aerosols composition is shown in Table 1. Average concentrations in our study were in the same order of magnitude, than previous measurements collected in the same region and season (Becagli et al., 2012; Calzolai et al., 2015; Tovar-Sánchez et al., 2014), and hence were consistent with Western Mediterranean background concentrations. No clear gradient (North-South or East-West) in atmospheric metals concentrations was observed during the cruise. Here, metal aerosols composition was mainly influenced by air masses from North of Europe and Atlantic Ocean (Figure S1), except between June 1st and June 4th, i.e for the stations St 9 and Fast 1-4 where African air masses loaded by dust were observed (Figure S1-2). During this period, the aerosol mass concentrations were the highest observed during the cruise with a maximum around 25 µg.m$^{-3}$, nevertheless these concentrations were typical of a moderate dust event (Pey et al., 2013). Aerosol Fe concentrations were the highest measured during the cruise, in average 245 ng.m$^{-3}$ during this period. The same observation was done for Co. A good correlation between Ni and V in the collected aerosols all along the cruise, suggest a common source associated to heavy oil combustion; i.e. marine ship traffic (Becagli et al., 2012). Some rains occurred during the cruise, but only one was measured when the vessel was in station, June 5th from 2:36 am to 3:04 am between Fast 3 and Fast 4 samples. However, all the zone around the Fast station was rainy from the 3rd of June (Figure S3). As the collected rain composition was typical to dust wet deposition with high particulate concentrations of Al, Fe and Ca (Fu et al., in preparation), we suppose the rain-out of dust in the atmospheric column around this station occurred between the 3rd to the 5th of June.

### 3.2. Biochemical composition and distribution of the surface water

Trace metals concentrations in the surface waters of the MS varied depending on the compartmentation (i.e. SML or SSW) and along a longitudinal gradient (Table 1).

#### 3.2.1. Trace metals in the SML

Trace metals concentrations of T-SML (Table 1) were lower, although, with the exception of Pb, of the same order of magnitude than those measured in the previous studies carried out in the MS (Tovar-





Sánchez et al., 2014). This lower content of metals in the SML is likely related with the lower dust aerosol's deposition during our sampling period and by the lack of influence of desert dust aerosols except for Fast station. T-SML concentration of Pb (average 663 ± 320 pM) was one order of magnitude lower than in previous studies impacted by pulses of both mineral African dust and anthropogenic aerosols coming from Europe (5596 ± 1589 pM) (Tovar-Sánchez et al., 2014). The highest concentrations of some metals were measured at the station FAST-3 (Co: 773,6 pM; Cu: 20.1 nM; Fe: 1433.3 nM and Pb: 1294.7 pM), likely impacted by the dusty rain events on this area.

Dissolved concentrations of Co, Zn, Pb, Cu and Ni showed a decreasing trend from the SML to the SSW, with concentrations 10.4 ± 0.7; 9.3 ± 5.5; 4.2 ± 1.8; 3.1 ± 1.5; and 1.2 ± 0.1 times higher in the SML than in the SSW, respectively. Vanadium (1.2 ± 0.42) and Fe (1.3 ± 1.5) varied lightly between SML and underlayer water, and Mo (1.0 ± 0.1) did not showed any differences (Table 1). Only Cd concentrations were consistently lower in the SML compared to the underlayer water (0.8 ± 0.2 times lower). Such depletion of dissolved metals in the SML compared to the underlayer water has been previously observed in areas with no significant aerosols inputs (Ebling and Landing, 2015, 2017). Although not fully understood, some mechanisms such as dominance of removal mechanisms versus diffusion, or higher influence of underlaying metal sources have been previously suggested to explain this metal depletion (Ebling and Landing, 2017; Hunter, 1980).

Spatial distribution of Co and Ni concentrations in the D-SML were well correlated with those measured in the D-SSW (Spearman's correlation coefficient ($r_s$): 0.87 for Co and 0.91 for Ni; p<0.01, Table 2), indicating for these elements an efficient diffusive mixing between these two compartments. These elements were also well correlated with the surface salinity distribution ($r_s$: 0.62 for Co and 0.93 for Ni; p<0.01, Table 2), and presented an eastward trend of increasing concentration, which is consistent with the characteristic distribution of metals on the surface of the MS (see section 3.2.4. below). Variations in concentrations for the rest of the elements (i.e. Cd, Cu, Fe, Pb, V and Zn) in the D-SML were not correlated either with the underlayer water or salinity gradient. Multiple physical, chemical and biological processes taking place in the SML could be affecting and controlling the mobility and diffusion of these elements between compartments. However, the concentrations of Cu, Fe and Zn in the T-SML showed an opposite trend with a longitudinal gradient inversely correlated with the salinity ($r_s$: -0.59 for





Cu; -0.69 for Fe, and -0.61 for Zn; p<0.01, Table 2). Since aerosols metal concentrations did not show any longitudinal trend and no other natural or anthropogenic sources were identified in the region, gradient concentration of these reactive trace elements in the T-SML must be influenced by other factors different than sources inputs, water exchange or dilution with Atlantic waters.

### 3.2.2. Residence time of trace metals in the SML

Estimates of the residence times of metals from aerosols inputs in the SML are critical to better understand the biogeochemical processes that affect the fate and distribution of trace metals in the surface ocean. To estimate the residence time ($t$) of particulate metals in the SML, we used the equation proposed by Ebling and Landing (2017) (Ebling and Landing, 2017):

$t = [TE]_{SML} \times d / J_{aerosol}$

where $[TE]_{SML}$ is the concentration of trace element (TE) in the T-SML, $d$ is the thickness of the SML and, and $J_{aerosol}$ is the measured aerosol trace metal flux. $J_{aerosol}$ was estimated by multiplying the metals aerosol concentrations with the sedimentation velocity, which is dependent on aerosols size. Thus, we used a velocity of mineral dust deposition for Fe (1 cm.s$^{-1}$) and an average velocity of fine anthropogenic particles for the other metals, i.e. 0.1 cm.s$^{-1}$ (Baker et al., 2010; Duce et al., 1991). For the calculations of the residence times along our different stations we used simultaneous empirical measurements of metals concentrations in the SML and metals aerosol fluxes (Table 3). Residence time of particulate metals (T-SML) ranged from 12 min for Co and Fe to 7.6 h for Cu. Since Mo and Cd are not enriched in the SML they were not considered in this calculation. Although variable among stations, residence time of Cu (3.0 ± 1.9 h), Zn (2.1 ± 0.8 h), V (1.7 ± 0.4 h), Pb (1.4 ± 0.7 h), Ni (55 ± 14 min), Fe (5.0 ± 3.1 min, excluding the station Fast 3) and Co (2.7 ± 0.9 min, excluding the station Fast 3) were relatively consistent with previous estimates in regions under low aerosols inputs (Ebling and Landing, 2017). Our results indicate that Fast 3 station was affected by the dusty rain events, which increased the concentration of some metals in the T-SML. Iron and Co were the elements that increased the most with major impact on their residence time for that station, i.e. multiplied by a factor 10 (Table 3). On average, while the highest residence times obtained for Cu, Pb and Zn are in agreement with their strong affinity to particles and therefore with a high probability of retention in the SML, other reactive elements such as Co and Fe presented, the shortest





residence time. Since, such fast transfer of these particle metals to the underlayer water (in the order of 1-3 min) is unlikely (mainly due their affinities to organic ligands), and dissolution is not immediately reflected in an increase of concentration in the dissolved fraction (i.e. D-SML), other parameters (linked to dynamic or biological activity) would be affecting the residence time of these elements in the SML. In the case of Co, wind seems to partly explain this short residence time. Wind speed showed high and significant negative correlation with the residence time of Co ($r_s$: -0.67, $p<0.01$), Ni ($r_s$: -0.76 $p<$ $p<0.01$) and V ($r_s$: -0.80 $p<$ $p<0.01$) in the SML, suggesting an influence of wind on the diffusion of these elements to the underlayer water (Table 2).

### 3.2.3. Neuston composition

The microbial composition (Bacteria; High nucleid acid-content bacteria: HNA; Low nucleid acid-content bacteria: LNA; phytoplankton; phytoplankton small; phytoplankton middle; phytoplankton large; Small cyanobacteria like cells: CBL-small; middle-large cyanobacteria like cells: CBL-middle-large) and abundance in the SML are shown in Table 1. Bacterial abundance in the SML ranged from $2x10^5$ to $1x10^6$ cell mL$^{-1}$ (average: $5.1x10^5 \pm 2.2x10^5$ cell mL$^{-1}$) that is of the same order of magnitude than abundance measured in the SML of other regions (e.g. in the Peruvian Coast with average of $8.9x10^5$ $\pm 4.3x10^5$ cell mL$^{-1}$) (Zäncker et al., 2018). Bacterial community was dominated by low nucleid acid-content bacteria (LNA) with an average concentration of $2.8 x10^5 \pm 1.0x10^5$ cell mL$^{-1}$. In general, and with the exception of phytoplankton middle and CBL-small, microbial abundance was higher in the SML than in the SSW with abundances ranging from 1 to 6 times higher for bacteria and CBL-middle-large, respectively (Table 1).

A microbial abundance decreases from west to east related to the increasing oligotrophy (explained though an increased P limitation) of the surface Mediterranean waters has been described (Pulido-Villena E. et al., 2012). In this study, microbial abundance in the SML and T-SML reactive elements (i.e. Cu, Fe, and Zn) showed the same longitudinal gradients with decreasing eastward concentration along the southern coast of the MS. In fact, bacterial abundances were significantly and positively correlated with these bioactive T-SML metals (i.e. $r_s$: Cu: 0.65 $p<0.01$; Fe: 0.53 $p<0.05$; and Zn: 0.49 $p<0.05$), suggesting that bacterioneuston could be affecting the concentration and fate of Cu, Fe





and Zn in the SML. Bacteria would efficiently assimilate the available fraction of Cu, Fe and Zn, resulting in a decrease and increase of the D-SML and T-SML fractions, respectively (Table 1). No general relationship between concentrations of metals and TEPs (high molecular weight polymers released by phytoplankton and bacteria and with high metal binding capacity (Passow, 2002)) were found in the SML,

although both showed strongly increased values after a dust deposition event at FAST 3 (Table 4). We therefore assume that metal assimilation by microbial communities would also explain the higher residence time of Cu and Zn (in the order of hours) in the SML. However, in the case of Fe with an estimated residence time of a few minutes, other processes different than wind speed and neuston uptake, should be contributing to facilitate the transfer from the SML to the underlayer water. For example,

photochemical reactions drive by intense solar radiation exposure in the SML could play an important role in the dissolution processes of this metal (Boyd et al., 2010). On the other hand, Ni was strongly and negatively correlated with bacteria abundance in the D-SML ($r_s$ = -0.93, p<0.01; $R^2$ = 0.74, p<0.01) suggesting, contrarily to Cu, Fe and Zn, a possible inhibiting role on the microbiology growth (Table 4 and Figure 2) (see next section for more discussion).

### 3.2.4. Subsurface water

The D-SSW concentrations of Cd, Co, Cu, Ni, Mo and Zn showed a longitudinal gradient of concentrations increasing from west to east, with strong significant positive correlations with longitude for Cd, Co and Ni (Figure 3). This trend is consistent with previous studies where the increased eastward

concentration along the southern coast of the MS is indicated to be due to factors such as: more intense Saharan deposition on the eastern MS (Guieu et al., 2002); more rapid exchange of water masses and margin inputs in the Western part (Yoon et al., 1999) or, as suggested for Co, the regeneration of biogenic particulate eastward that yields a westward decreased of the dissolved Co in surface (Dulaquais et al., 2017). Since surface salinity showed the same eastward increase and was close correlated with those

metals ($r_s$ ranged from 0.51 p<0.05 for Mo to 0.97 p<0.01 for Ni; Table 2), the exchange with the surface Atlantic Ocean waters seems to be the main cause of this gradient of concentrations in our study, although higher aerosol inputs in the western MS could also contribute to this gradient. Other metals (i.e. Fe, Pb and V) did not show any clear geographical trend and variations in surface concentrations could be





influenced by several factors other than dilution or exchange, such as vertical diffusive fluxes or, specific metal sources, as it is in the case of Fe and Pb that has been suggested to be more affected by atmospheric inputs (Nicolas et al., 1994; Yoon et al., 1999). In fact, Pb was the only element that showed significant positive correlation with latitude ($r_s$: 0.88 p<0.01, Table 2) suggesting an influence of the northern region

of the MS.

D-SSW concentrations of Ni were strongly and negatively correlated with microbial abundance (mainly with heterotrophic bacterial) in the underlaying water ($r_s$ : -0.91, p<0.01; $R^2 = 0.87$, p<0.01) (Figure 2 and Table 4) suggesting a potential negative role of this metal on bacteria and small phytoplankton on the top meter of the surface western MS, including the SML. Toxicity by Ni at

concentrations of ~50 nM has been previously demonstrated in the western MS, with inhibitions of 10% ($EC_{10}$) in phycoerythrin and Chl-a signals of natural population of the picoplankter *Synechococcus* sp. (Debelius et al., 2011). Although that toxicity concentration (tested in picoplankton), is around 13 time higher than average values measured in our samples (T-SML: 4.1 ± 0.5 nM, D-SML: 3.9 ± 0.6 nM and D-SSW: 3.2 ± 0.6 nM; Table 1), deleterious effects on the neuston and microbial communities to lower

concentrations could be feasible in the top meter of the surface sea. Indeed, UV radiations in this surface layer are highly intense and can acts as a biochemical microreactor where many transformations and photochemical reaction occurs (Wurl et al., 2017) affecting complexation, solubilization, bioaccumulation and/or toxicity of many trace elements. Even if a general decreasing trend from west to east of microbial abundance due to the increasing oligotrophy has been described, it is interesting to

mention that primary production and Chl-a concentration (measured a 5 m depth), did not show significant correlations with Ni (Table 4). We therefore assume that toxicity of Ni is mainly affecting the bacterial community and/or on the top meter of the surface ocean. Nickel, as other transition metals, is an essential cofactor of several enzymes, however, it becomes toxic when homeostasis fails. Multiple potential mechanisms of Ni toxicity to aquatic organisms, and in particular to bacteria, have been identified

(Macomber and Hausinger, 2016). Among the different possible toxicity mechanisms (including inhibition of Zn and Fe metalloenzymes and non-metalloenzymes) the toxicity involving reactive oxygen species (ROS) is likely the most feasible in surface seawater. While Ni itself is a poor generator of reactive oxygen species (ROS) when compared to other metals like Fe or Cu, its reactivity and ROS productions





can be enhanced by the displacement of redox-active iron from iron metallocenters (Macomber and Hausinger, 2016) or when chelated by oligopeptides and histidine (Brix et al., 2017), which are abundant in the SML. It appears that Nickel-dependent toxicity involving ROS may be likely mechanism of oxidative stress in marine microbial organism of the surface ocean. There is many information on the
5 effect of Ni on insolate laboratory microalgae experiments, however its toxicity role in oceans have been poorly explored. Therefore, additional studies on Ni diffusión from SML, solubility, speciation, and the effects on phytoplankton at the species level are required to fully understand the magnitude of this finding.

## 4. Conclusions

Our results show that the SML in the MS is enriched in trace metals relative to the SSW even under low aerosols deposition rates. While some metals entering in the SML (e.g. Cd, Co, Ni and V) show efficient diffusive mixing from SML to SSW, other more reactive metals such as Cu, Fe, Pb and Zn affected by chemical and biological processes show a major difficulty of mobility. A strong negative correlation between Ni concentration and heterotrophic bacterial abundance in the SML and SSW suggest an
inhibiting role of this element on the microbial growth in the top meter of the surface.

## Acknowledgements

This work is a contribution of the project MEGOCA - CTM2014-59244-C3-3-R (Spain); and the
20 PEACETIME project (http://peacetime-project.org), a joint initiative of the MERMEX and ChArMEx components supported by CNRS-INSU, IFREMER, CEA, and Météo-France as part of the programme MISTRALS coordinated by INSU (doi: 10.17600/17000300). All data have been acquired during the PEACETIME oceanographic expedition on board R/V Pourquoi Pas? in May-June 2017; The authors thank I. Carribero and J. Pampin for their assistance with chemical analysis. We thank Julia Uitz, Céline
Dimier and the SAPIGH analytical platform at IMEV for HPLC for sampling and analyses of Chl-a. We thank the crews of the R/V Pourquoi Pas? for technical assistance on the field. A. Rodríguez-Romero is supported by the Spanish grant Juan de la Cierva Formación 2015" (JCI-2015-26873). M.B was funded



from the European Union Seventh Framework Program ([FP7/2007-2013]) under grant agreement no. [PIOF-GA-2012-626734] (IRON-IC project).

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



**Figure 1.** Location of the sampling stations in the studied region.



**Figure 2.** Concentration of Dissolved Ni plotted against bacterial abundance in the SML (black dots) and in the Subsurface water (red open dots). The lines represent linear regression equations.





**Figure 3.** Salinity and dissolved trace metals concentration in the subsurface waters plotted against longitude. The dashed lines represent linear regression equations.







**Table 1.** Parameters measured in all stations. Metals in pmol.m⁻³ or nmol.m⁻³ for aerosols and nM or pM in water compartments.

| Station | Date | Time UTC | Lat | Long | Wind Speed knots | Salinity 5m depth PSU | Temp 5m depth °C | Sample | Cd nM/ng·m⁻³ | Co pM/ng·m⁻³ | Cu nM/ng·m⁻³ | Fe nM/ng·m⁻³ | Mo nM/ng·m⁻³ | Ni nM/ng·m⁻³ | Pb pM/ng·m⁻³ | V nM/ng·m⁻³ | Zn nM/ng·m⁻³ | Bacteria cell·ml⁻¹ x10⁵ | Bacteria HNA x10⁵ | Bacteria LNA x10⁵ | Phyto cell·ml⁻¹ x10² | Phyto small x10² | Phyto middle x10² | Phyto large cell·ml⁻¹ | CBL small cell·ml⁻¹ | CBL middle-large cell·ml⁻¹ | TEP particles·l⁻¹ x10⁶ | PP mgC·m⁻³·d⁻¹ | Chl-a µg·l⁻¹ |
|---|---|---|---|---|---|---|---|---|---|---|---|---|---|---|---|---|---|---|---|---|---|---|---|---|---|---|---|---|---|
| St. 1 | 12/5/17 | 13:15 | 41.8918 | 6.3333 | 18.8 | 38.2 | 16.4 | Aerosol | 0.1 | 0.1 | 2.0 | 37.9 | 0.4 | 2.7 | 1.1 | 8.0 | 3.6 | | | | | | | | | | | | |
| | | | | | | | | D-SML | 74.0 | 109.8 | 8.0 | 6.8 | 125.9 | 4.1 | 323.0 | 14.9 | 8.0 | 3.49 | 1.62 | 1.88 | 84.71 | 7.51 | 29.85 | 3.43 | 3685.32 | 1080.89 | 7.67 | 5.555 | |
| | | | | | | | | T-SML | 76.3 | 112.6 | 3.6 | 81.8 | 121.0 | 4.5 | 1217.5 | 21.7 | 11.1 | 3.02 | 1.26 | 1.78 | 70.74 | 8.34 | 33.70 | 6.86 | 2707.37 | 192.16 | 4.79 | | 0.187 |
| | | | | | | | | SSW | 78.5 | 11.3 | 1.6 | 14.3 | 130.1 | 3.5 | 111.0 | 23.0 | 1.9 | | | | | | | | | | | | |
| St. 3 | 14/5/17 | 5:30 | 39.1333 | 7.6835 | 5.6 | 37.2 | 18.7 | Aerosol | 0.0 | 0.1 | 0.4 | 22.9 | 0.0 | 2.4 | 0.3 | 6.5 | 3.3 | | | | | | | | | | | | |
| | | | | | | | | D-SML | 62.7 | 159.6 | 5.8 | 3.7 | 132.5 | 4.0 | 766.0 | 10.6 | 8.5 | 5.06 | 1.80 | 3.30 | 30.76 | 16.88 | 5.78 | 3.44 | 10.32 | 832.13 | 8.24 | 1.613 | |
| | | | | | | | | T-SML | 72.5 | 163.1 | 7.5 | 72.2 | 124.9 | 4.0 | 858.2 | 25.4 | 12.4 | 5.68 | 2.10 | 3.62 | 21.72 | 14.89 | 5.88 | 0.00 | 17.19 | 113.47 | 1.50 | | 0.095 |
| | | | | | | | | SSW | 69.3 | 13.6 | 1.2 | 2.3 | 124.3 | 2.6 | 83.6 | 23.8 | 0.6 | | | | | | | | | | | | |
| St. 4 | 15/5/17 | 7:30 | 37.9832 | 7.9768 | 6.8 | 37.1 | 19.8 | Aerosol | 0.0 | 0.1 | 0.5 | 27.3 | 0.1 | 2.9 | 0.2 | 6.8 | 2.6 | | | | | | | | | | | | |
| | | | | | | | | D-SML | 57.0 | 143.3 | 5.5 | 1.5 | 121.7 | 3.6 | 601.5 | 15.2 | 14.7 | 6.92 | 3.58 | 3.37 | 31.11 | 17.71 | 5.28 | 0.00 | 10.29 | 837.26 | 5.97 | 1.709 | |
| | | | | | | | | T-SML | 59.7 | 172.0 | 12.5 | 32.6 | 122.4 | 4.0 | 881.3 | 27.1 | 12.6 | 6.65 | 3.59 | 3.10 | 24.35 | 16.02 | 7.14 | 0.00 | 17.16 | 137.26 | 2.66 | | 0.090 |
| | | | | | | | | SSW | 65.0 | 15.1 | 7.2 | 5.4 | 125.0 | 2.7 | 87.4 | 18.4 | 0.7 | | | | | | | | | | | | |
| St. 5 | 16/5/01 | 6:00 | 38.9532 | 11.023 | 15.5 | 37.8 | 19.9 | Aerosol | 0.0 | 0.1 | 0.6 | 57.3 | 0.4 | 1.9 | 0.7 | 3.9 | 2.6 | | | | | | | | | | | | |
| | | | | | | | | D-SML | 66.0 | 143.0 | 2.6 | 0.6 | 122.6 | 4.0 | 212.7 | 35.0 | 4.7 | 4.27 | 2.05 | 2.25 | 16.76 | 8.77 | 3.37 | 0.00 | 3.44 | 495.15 | 4.55 | 1.642 | |
| | | | | | | | | T-SML | 64.4 | 152.9 | 4.8 | 37.4 | 128.6 | 3.9 | 559.3 | 31.1 | 11.1 | 4.45 | 2.12 | 2.36 | 13.74 | 7.98 | 5.02 | 0.00 | 17.19 | 92.84 | 2.28 | | 0.060 |
| | | | | | | | | SSW | n.m | 13.8 | 1.8 | 2.0 | 130.7 | 3.2 | 101.2 | 18.7 | 1.2 | | | | | | | | | | | | |
| TYR-1 | 16/5/01 | 8:00 | 39.34 | 12.593 | 6.6 | 37.8 | 20.3 | Aerosol | 0.1 | 0.1 | 4.8 | 115.3 | 0.3 | 3.1 | 1.9 | 5.6 | 7.2 | | | | | | | | | | | | |
| | | | | | | | | D-SML | 59.9 | 149.7 | 4.3 | 4.4 | 74.8 | 3.8 | 293.4 | 32.2 | 6.7 | 4.84 | 2.00 | 3.46 | 16.18 | 5.90 | 3.95 | 6.86 | 20.59 | 641.67 | 3.55 | 1.518 | |
| | | | | | | | | T-SML | 59.1 | 145.3 | 11.7 | 55.2 | 119.1 | 3.8 | 488.3 | 24.6 | 7.6 | 4.89 | 1.87 | 3.46 | 12.88 | 6.26 | 4.37 | 13.75 | 37.82 | 209.75 | 0.68 | | 0.063 |
| | | | | | | | | SSW | 75.1 | 13.3 | 1.7 | 2.1 | 122.9 | 3.2 | 90.1 | 19.4 | 1.0 | | | | | | | | | | | | |
| TYR-2 | 18/5/17 | 8:00 | 39.3398 | 12.593 | 4.9 | 37.7 | 21.1 | Aerosol | 0.1 | 0.1 | 2.3 | 162.3 | 0.1 | 2.0 | 3.9 | 3.9 | 8.6 | | | | | | | | | | | | |
| | | | | | | | | D-SML | 65.7 | 143.6 | 7.9 | 1.1 | 137.7 | 3.9 | 316.3 | 29.8 | 7.7 | 5.53 | 2.00 | 3.57 | 34.60 | 7.08 | 7.58 | 17.87 | 82.21 | 1930.04 | 15.10 | 1.725 | |
| | | | | | | | | T-SML | 79.9 | 160.6 | 5.3 | 82.5 | 124.9 | 3.6 | 435.2 | 24.4 | 7.9 | 5.00 | 1.87 | 3.16 | 17.26 | 6.61 | 8.15 | 14.30 | 92.93 | 178.71 | 1.84 | | 0.071 |
| | | | | | | | | SSW | 74.5 | 14.1 | 2.0 | 2.1 | 130.0 | 3.2 | 90.7 | 18.6 | 1.0 | | | | | | | | | | | | |
| St. 6 | 22/5/17 | 6:00 | 38.8077 | 14.5 | 10.1 | 37.4 | 20.4 | Aerosol | 0.1 | 0.1 | 4.8 | 189.8 | 1.2 | 2.2 | 2.7 | 2.9 | 11.3 | | | | | | | | | | | | |
| | | | | | | | | D-SML | 55.5 | 169.7 | 2.7 | 13.7 | 131.7 | 3.7 | 281.7 | 25.2 | 4.2 | 5.19 | 2.17 | 3.04 | 20.61 | 9.84 | 3.53 | 3.60 | 0.00 | 756.58 | 8.44 | n.m | |
| | | | | | | | | T-SML | 56.8 | 144.2 | 4.4 | 42.0 | 116.6 | 3.5 | 572.2 | 23.2 | 9.2 | 4.99 | 2.03 | 3.00 | 12.78 | 7.83 | 4.15 | 3.61 | 0.00 | 111.92 | 7.47 | n.m | n.m |
| | | | | | | | | SSW | 72.9 | 15.7 | 1.7 | 2.3 | 131.2 | 3.1 | 86.1 | 18.4 | 1.0 | | | | | | | | | | | | n.m |
| SAV | 23/5/17 | 10:00 | 37.8401 | 17.602 | 3.0 | 38.5 | 20.1 | Aerosol | 0.1 | 0.1 | 2.1 | 89.9 | 2.1 | 7.1 | 1.4 | 18.5 | 4.9 | | | | | | | | | | | | |
| | | | | | | | | D-SML | 77.5 | 158.9 | 2.9 | 2.2 | 129.8 | 4.7 | 258.1 | 25.9 | 6.6 | 3.66 | 1.37 | 2.30 | 13.76 | 5.01 | 3.82 | 14.41 | 18.01 | 497.18 | 19.10 | n.m | |
| | | | | | | | | T-SML | 67.4 | 171.4 | 5.0 | 26.7 | 132.3 | 4.9 | 1028.8 | 27.1 | 7.9 | 3.53 | 1.30 | 2.25 | 12.83 | 6.23 | 5.73 | 3.60 | 14.41 | 104.48 | 3.34 | | n.m |
| | | | | | | | | SSW | 80.8 | 15.4 | 1.6 | 2.1 | 129.3 | 4.3 | 59.1 | 20.4 | 1.1 | | | | | | | | | | | | |
| St. 7 | 24/5/17 | 4:00 | 36.6035 | 18.166 | 4.8 | 38.5 | 20.8 | Aerosol | 0.0 | 0.1 | 2.5 | 85.6 | 0.1 | 4.2 | 2.5 | 8.7 | 6.9 | | | | | | | | | | | | |
| | | | | | | | | D-SML | 74.1 | 176.1 | 6.0 | 1.6 | 129.0 | 4.7 | 222.0 | 24.8 | 9.7 | 3.08 | 1.28 | 1.81 | 20.38 | 5.79 | 3.76 | 6.89 | 172.28 | 940.67 | 8.36 | n.m | |
| | | | | | | | | T-SML | 94.1 | 187.6 | 8.3 | 48.6 | 121.4 | 4.6 | 392.3 | 25.2 | 10.7 | 3.16 | 1.32 | 1.85 | 15.63 | 4.44 | 4.93 | 3.45 | 379.02 | 279.10 | 2.96 | 0.873 | |
| | | | | | | | | SSW | 74.0 | 15.8 | 1.7 | 2.9 | 126.2 | 3.9 | 60.1 | 19.7 | 1.0 | | | | | | | | | | | | 0.056 |
| ION-1 | 25/5/17 | 8:00 | 35.4892 | 19.776 | 8.1 | 38.8 | 21.0 | Aerosol | 0.2 | 0.1 | 1.2 | 88.8 | 0.2 | 7.1 | 1.8 | 17.9 | 11.0 | | | | | | | | | | | | |
| | | | | | | | | D-SML | 53.4 | 188.8 | 3.3 | 5.4 | 91.7 | 5.1 | 284.6 | 38.2 | 9.9 | 2.11 | 0.81 | 1.30 | 17.08 | 5.34 | 3.03 | 13.78 | 68.91 | 823.51 | 26.55 | 1.790 | |
| | | | | | | | | T-SML | 69.8 | 151.5 | 6.8 | 24.7 | 108.4 | 4.6 | 527.9 | 23.9 | 9.1 | 2.19 | 0.88 | 1.32 | 14.29 | 4.71 | 5.88 | 6.88 | 151.29 | 247.57 | 8.20 | | n.m |
| | | | | | | | | SSW | 93.6 | 17.1 | 1.9 | 2.0 | 128.5 | 4.1 | 61.4 | 18.7 | 1.2 | | | | | | | | | | | | |
| ION-2 | 27/5/17 | 8:00 | 35.4892 | 19.777 | 12.4 | 38.8 | 21.1 | Aerosol | 0.0 | 0.0 | 2.9 | 80.6 | 0.1 | 1.7 | 2.5 | 3.8 | 11.3 | | | | | | | | | | | | |
| | | | | | | | | D-SML | 13.1 | 154.9 | 2.8 | 4.1 | 130.4 | 4.4 | 74.8 | 19.0 | 7.1 | 2.04 | 0.94 | 1.10 | 19.14 | 4.02 | 5.33 | 6.88 | 189.12 | 818.36 | 13.27 | 1.745 | |
| | | | | | | | | T-SML | 13.7 | 168.1 | 5.9 | 30.7 | 127.1 | 4.7 | 74.8 | 28.3 | 13.6 | 2.16 | 0.96 | 1.21 | 9.50 | 0.00 | 4.00 | 0.00 | 275.65 | 310.11 | 7.16 | | 0.063 |
| | | | | | | | | SSW | 85.2 | 16.7 | 2.3 | 2.1 | 128.9 | 4.0 | 61.9 | 22.4 | 1.1 | | | | | | | | | | | | |
| St. 8 | 30/5/17 | 5:00 | 36.2103 | 16.631 | 3.7 | 37.9 | 21.2 | Aerosol | 0.0 | 0.1 | 2.4 | 89.8 | 0.5 | 4.4 | 1.1 | 11.2 | 12.5 | | | | | | | | | | | | |
| | | | | | | | | D-SML | 62.9 | 171.5 | 4.4 | 0.2 | 137.5 | 4.3 | 301.1 | 15.5 | 4.8 | 3.70 | 1.62 | 2.10 | 30.59 | 6.22 | 5.95 | 58.45 | 130.66 | 1688.30 | 26.13 | 1.604 | |
| | | | | | | | | T-SML | 69.6 | 163.1 | 3.5 | 13.9 | 124.8 | 3.8 | 404.0 | 29.4 | 4.1 | 3.24 | 1.48 | 1.77 | 11.64 | 4.95 | 4.09 | 3.44 | 120.35 | 171.93 | 4.65 | | 0.070 |
| | | | | | | | | SSW | 76.7 | 16.5 | 1.5 | 3.3 | 123.8 | 3.5 | 62.8 | 16.3 | 1.1 | | | | | | | | | | | | |
| St. 9 | 31/5/17 | 16:00 | 38.1347 | 5.8408 | 6.2 | 37.1 | 22.0 | Aerosol | 0.1 | 0.1 | 1.8 | 170.7 | 0.2 | 2.9 | 5.7 | 4.1 | 10.2 | | | | | | | | | | | | |
| | | | | | | | | D-SML | 51.0 | 145.9 | 5.4 | 0.9 | 135.2 | 3.9 | 373.6 | 22.5 | 21.1 | 5.88 | 3.01 | 2.91 | 33.82 | 9.32 | 11.76 | 30.95 | 30.95 | 1248.18 | 37.07 | 2.570 | |
| | | | | | | | | T-SML | 60.7 | 149.9 | 7.9 | 82.5 | 120.8 | 3.6 | 596.5 | 21.9 | 11.6 | 5.65 | 3.18 | 2.51 | 16.70 | 8.56 | 7.29 | 0.00 | 34.39 | 85.96 | 10.57 | | 0.072 |
| | | | | | | | | SSW | 66.9 | 14.5 | 1.8 | 2.0 | 130.5 | 3.0 | 91.8 | 21.2 | 1.0 | | | | | | | | | | | | |
| FAST-1 | 2/6/17 | 17:00 | 37.946 | 2.902 | 13.8 | 36.7 | 21.7 | Aerosol | 0.0 | 0.1 | 4.6 | 226.9 | 0.5 | 5.5 | 7.0 | 4.7 | 14.6 | | | | | | | | | | | | |
| | | | | | | | | D-SML | 46.6 | 118.5 | 3.8 | 0.1 | 125.4 | 3.4 | 359.1 | 24.0 | 11.9 | 6.20 | 3.55 | 2.68 | 17.11 | 9.54 | 5.07 | 0.00 | 3.45 | 282.54 | 6.93 | 2.612 | |
| | | | | | | | | T-SML | 45.0 | 408.4 | 17.2 | 207.4 | 124.7 | 4.4 | 880.5 | 26.0 | 19.6 | 6.09 | 3.76 | 2.38 | 14.53 | 9.06 | 4.93 | 0.00 | 3.45 | 86.14 | 1.27 | | 0.070 |
| | | | | | | | | SSW | 57.3 | 12.0 | 1.1 | 2.8 | 114.3 | 2.5 | 74.4 | 17.5 | 0.7 | | | | | | | | | | | | |
| FAST-3 | 4/6/17 | 6:00 | 37.947 | 2.91153 | 4.9 | 36.7 | 21.7 | Aerosol | 0.1 | 0.2 | 0.9 | 245.3 | 0.2 | 4.5 | 3.4 | 10.2 | 6.8 | | | | | | | | | | | | |
| | | | | | | | | D-SML | 47.7 | 142.8 | 5.2 | 4.3 | 124.2 | 3.2 | 454.6 | 25.4 | 8.9 | 9.99 | 5.42 | 4.59 | 44.31 | 23.14 | 7.01 | 6.88 | 799.51 | 1437.29 | 37.29 | 2.378 | |
| | | | | | | | | T-SML | 58.5 | 773.6 | 20.1 | 1433.3 | 101.9 | 4.5 | 1294.7 | 26.1 | 12.1 | 6.35 | 3.77 | 2.62 | 18.35 | 12.62 | 4.71 | 0.00 | 2186.27 | 120.35 | 2.16 | | 0.085 |
| | | | | | | | | SSW | 57.0 | 12.5 | 1.5 | 1.6 | 120.5 | 2.6 | 74.7 | 19.6 | 1.1 | | | | | | | | | | | | |
| FAST-4 | 5/6/17 | 8:00 | 37.9467 | 2.9168 | 8.7 | 36.6 | 21.8 | Aerosol | 0.2 | 0.3 | 1.2 | 266.0 | 0.2 | 6.8 | 4.3 | 17.1 | 8.3 | | | | | | | | | | | | |
| | | | | | | | | D-SML | 53.6 | 115.9 | 4.5 | 3.4 | 125.4 | 2.9 | 247.3 | 17.5 | 20.5 | 8.80 | 4.01 | 4.83 | 44.89 | 18.68 | 7.54 | 10.61 | 799.51 | 1093.13 | n.m | 2.778 | |
| | | | | | | | | T-SML | 57.5 | 96.8 | 17.0 | 64.1 | 116.9 | 3.2 | 635.4 | 20.5 | 0.6 | 6.87 | 3.41 | 3.49 | 43.40 | 12.74 | 7.11 | 3.54 | 2186.27 | 201.65 | n.m | | 0.081 |
| | | | | | | | | SSW | 64.4 | 11.9 | 0.8 | 4.0 | 113.0 | 2.5 | 73.2 | 18.4 | 2.0 | | | | | | | | | | | | |
| FAST-5 | 7/6/01 | 5:30 | 37.9465 | 2.9167 | 18.2 | 36.6 | 21.9 | Aerosol | 0.0 | 0.1 | 0.7 | 44.9 | 0.2 | 1.6 | 3.2 | 4.0 | 2.0 | | | | | | | | | | | | |
| | | | | | | | | D-SML | 45.0 | 122.7 | 4.6 | 0.0 | 130.3 | 2.9 | 283.3 | 18.9 | 8.3 | 6.61 | 3.22 | 3.41 | 25.04 | 13.76 | 4.39 | 7.08 | 113.20 | 604.94 | 3.14 | 1.993 | |
| | | | | | | | | T-SML | 53.3 | 122.7 | 13.6 | 31.1 | 127.3 | 3.4 | 426.1 | 28.6 | 17.8 | 6.64 | 3.57 | 3.10 | 23.54 | 14.05 | 5.68 | 3.53 | 201.23 | 211.82 | 2.88 | | 0.078 |
| | | | | | | | | SSW | 75.5 | 11.9 | 1.3 | 1.8 | 110.8 | 2.5 | 70.9 | 15.4 | 0.9 | | | | | | | | | | | | |

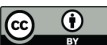



**Table 2.** Spearman's rank correlation coefficients for selected parameters. Significant correlations at p<0.05 and p<0.01 are marked with one asterisk (orange numbers) and two asterisks (red numbers), respectively.

| Group | Variable | Latitude | Longitude | Salinity | Wind Speed | T-SML Cd | Co | Cu | Fe | Mo | Ni | Pb | V | Zn | D-SML Cd | Co | Cu | Fe | Mo | Ni | Pb | V | Zn |
|---|---|---|---|---|---|---|---|---|---|---|---|---|---|---|---|---|---|---|---|---|---|---|---|
| | Latitude | 1 | -0.393 | -0.286 | 0.195 | 0.221 | -0.406 | -0.112 | .562* | -0.088 | -0.422 | 0.361 | -0.362 | -0.092 | 0.318 | -0.405 | 0.373 | 0.212 | -0.005 | -0.335 | 0.394 | -0.059 | -0.126 |
| | Longitude | -0.393 | 1 | .883** | -0.331 | 0.325 | 0.105 | -.589* | -.692** | 0.148 | 0.434 | -.560* | 0.136 | -.611** | 0.349 | .802** | -0.315 | 0.311 | 0.11 | .806** | -.535* | 0.325 | -0.27 |
| | Salinity | -0.286 | .883** | 1 | -0.2 | 0.455 | 0.127 | -.652** | -.530* | 0.161 | .670** | -0.362 | 0.098 | -.609** | 0.445 | .620** | -0.209 | 0.369 | -0.006 | .934** | -0.432 | 0.304 | -0.145 |
| D-SSW | Cd | -0.319 | .723** | .796** | 0.032 | 0.232 | -0.25 | -.682** | -.715** | 0.347 | 0.397 | -0.435 | 0.194 | -0.494 | 0.229 | 0.429 | -0.265 | 0.226 | 0.124 | .721** | -0.491 | 0.185 | -0.3 |
| | Co | -.575* | .857** | .672** | -0.427 | 0.186 | 0.338 | -0.426 | -.630** | 0.054 | 0.39 | -.495* | 0.228 | -0.478 | 0.086 | .870** | -0.346 | 0.103 | 0.196 | .685** | -0.301 | 0.277 | 0.015 |
| | Cu | -0.005 | .660** | .568* | 0.002 | 0.152 | 0.164 | -0.35 | -0.414 | 0.093 | 0.314 | -0.287 | 0.206 | -0.368 | 0.181 | 0.343 | -0.181 | 0.294 | -0.201 | 0.38 | -0.255 | 0.395 | -0.069 |
| | Fe | 0.171 | 0.002 | 0.007 | 0.012 | 0.226 | -0.072 | -0.184 | 0.058 | -0.08 | -0.048 | 0.128 | -0.318 | 0.055 | 0.387 | -0.112 | 0.372 | 0.067 | 0.088 | -0.006 | 0.135 | -.589* | -0.105 |
| | Mo | 0.278 | 0.446 | .507* | 0.036 | 0.348 | 0.157 | -.728** | -0.069 | 0.086 | 0.162 | -0.002 | -0.179 | -0.419 | 0.412 | 0.277 | -0.147 | 0.282 | 0.257 | .483* | -0.191 | 0.27 | -0.164 |
| | Ni | -0.307 | .883** | .970** | -0.341 | .486* | -0.145 | -.661** | -.532* | 0.163 | .634** | -0.327 | 0.086 | -.699** | .487* | .630** | -0.178 | 0.303 | 0.074 | .912** | -0.396 | 0.352 | -0.14 |
| | Pb | .884** | -0.435 | -0.298 | 0.364 | 0.096 | -0.365 | -0.137 | .547* | -0.125 | -0.424 | 0.306 | -0.287 | 0.017 | 0.115 | -.485* | 0.275 | 0.056 | -0.029 | -0.349 | 0.407 | -0.064 | 0.012 |
| | V | 0.228 | 0.231 | 0.459 | -0.153 | 0.359 | 0.058 | -0.221 | 0.234 | 0.015 | .525* | 0.197 | -0.248 | -0.094 | 0.262 | 0.173 | 0.208 | .511* | 0.093 | .518* | 0.047 | -0.043 | 0.131 |
| | Zn | 0.007 | 0.446 | .682** | -0.026 | 0.469 | -0.034 | -.677** | -0.24 | 0.139 | 0.395 | -0.098 | 0.107 | -.580* | 0.404 | 0.152 | -0.093 | 0.148 | 0.044 | .621** | -0.24 | 0.425 | -0.137 |
| Residence Time | Cd | 0.018 | 0.37 | 0.301 | -.757** | .816** | 0.218 | -0.127 | -0.088 | -0.113 | 0.162 | -0.056 | -0.167 | -.542* | .605* | .532* | .490* | 0.13 | 0.103 | 0.432 | 0.23 | 0.081 | 0.132 |
| | Co | -0.328 | 0.055 | 0.021 | -.674** | 0.136 | .901** | 0.359 | 0.161 | -0.061 | 0.463 | 0.188 | 0.227 | -0.113 | 0.064 | 0.289 | 0.144 | -0.128 | -0.082 | 0.107 | 0.358 | 0.13 | 0.429 |
| | Cu | -0.108 | -0.435 | -.558* | -0.206 | -0.262 | 0.282 | .934** | 0.338 | -0.382 | -0.113 | 0.203 | -0.108 | 0.417 | -0.373 | -0.213 | 0.255 | -0.083 | -0.436 | -.536* | 0.319 | -0.034 | 0.419 |
| | Fe | 0.448 | -.549* | -0.462 | -0.179 | 0.047 | 0.209 | .510* | .926** | -0.38 | -0.102 | .492* | -.502* | 0.258 | -0.069 | -0.356 | 0.458 | 0.129 | -0.018 | -0.407 | .538* | -0.011 | 0.322 |
| | Mo | -0.105 | 0.427 | 0.226 | -.847** | .490* | 0.328 | -0.091 | -0.255 | 0.179 | 0.127 | -0.083 | 0.11 | -0.431 | 0.456 | .527* | 0.201 | 0.005 | 0.221 | 0.318 | 0.118 | 0.007 | -0.076 |
| | Ni | -0.332 | 0.455 | 0.443 | -.764** | -0.078 | .542* | 0.054 | -0.211 | -0.137 | .613** | 0.115 | 0.029 | -0.385 | 0.368 | .527* | 0.221 | 0.284 | -0.164 | .508* | 0.137 | 0.118 | 0.24 |
| | Pb | 0.28 | -.546* | -0.415 | -0.056 | -0.078 | 0.221 | 0.299 | 0.426 | -0.181 | 0.14 | .975** | -0.154 | 0.255 | 0.044 | -0.4 | 0.093 | 0.15 | -0.299 | -0.347 | .561* | -0.13 | 0.225 |
| | V | -0.257 | 0.386 | 0.221 | -.798** | 0.309 | .537* | -0.011 | -0.371 | 0.115 | 0.277 | -0.039 | 0.415 | -0.409 | 0.336 | .490* | 0.08 | -0.005 | -0.006 | 0.262 | 0.13 | 0.013 | -0.042 |
| | Zn | -0.195 | -0.424 | -.547* | -0.004 | -0.284 | 0.262 | .826** | 0.297 | -0.159 | -0.012 | 0.25 | -0.11 | .755** | -0.466 | -0.255 | 0.299 | -0.083 | -0.194 | -0.448 | 0.289 | -0.338 | 0.473 |





**Table 3.** Residence times (hours) of particulate metals derived for aerosols deposition

| Satation1 | Co | Cu | Fe | Ni | Pb | V | Zn |
|---|---|---|---|---|---|---|---|
| **St. 1** | 0.02 | 0.8 | 0.09 | 0.7 | 1.6 | 1.0 | 1.5 |
| **St. 3** | 0.05 | 2.8 | 0.13 | 1.0 | 2.0 | 1.9 | 2.7 |
| **St. 4** | 0.06 | 4.7 | 0.06 | 1.1 | 2.0 | 2.1 | 2.7 |
| **St. 5** | 0.03 | 1.1 | 0.04 | 0.6 | 1.3 | 1.4 | 1.4 |
| **TYR-1** | 0.05 | 4.4 | 0.10 | 1.0 | 1.1 | 1.9 | 1.6 |
| **TYR-2** | 0.05 | 2.0 | 0.15 | 0.9 | 1.0 | 1.9 | 1.7 |
| **St. 6** | 0.03 | 1.2 | 0.06 | 0.7 | 1.3 | 1.3 | 1.5 |
| **SAV** | 0.06 | 1.9 | 0.05 | 1.3 | 2.4 | 2.1 | 1.7 |
| **St. 7** | 0.06 | 3.1 | 0.09 | 1.2 | 0.9 | 1.9 | 2.3 |
| **ION-1** | 0.05 | 2.6 | 0.04 | 1.2 | 1.2 | 1.8 | 2.0 |
| **ION-2** | 0.04 | 1.7 | 0.04 | 0.9 | 0.2 | 1.6 | 2.2 |
| **St. 8** | 0.05 | 1.3 | 0.02 | 1.0 | 0.9 | 2.2 | 0.9 |
| **St. 9** | 0.04 | 2.2 | 0.11 | 0.7 | 1.4 | 1.2 | 1.9 |
| **FAST-1** | 0.08 | 3.9 | 0.22 | 0.7 | 2.0 | 1.2 | 2.6 |
| **FAST-3** | 0.25 | 7.6 | 2.53 | 1.2 | 3.0 | 2.0 | 2.6 |
| **FAST-4** | 0.03 | 6.4 | 0.11 | 0.8 | 1.5 | 1.6 | 4.4 |
| **FAST-5** | 0.02 | 3.1 | 0.03 | 0.5 | 1.0 | 1.3 | 2.3 |
| Avergae | 0.06 | 3.00 | 0.23 | 0.92 | 1.45 | 1.68 | 2.11 |
| S.D. | 0.05 | 1.89 | 0.59 | 0.23 | 0.66 | 0.37 | 0.79 |




**Table 4.** Spearman's rank correlation coefficients for selected parameters. HNA: High nucleid acid-content bacteria;; HNA; LNA: Low nucleid acid-content bacteria; CBL-small: Small cyanobacteria like cells; CBL-middle-large: middle-high cyanobacteria like cells. Significant correlations at p<0.05 and p<0.01 are marked with one asterisk (orange numbers) and two asterisks (red numbers), respectively.

| Variable | SML Bacteria | SML HNA | SML LNA | SML Pico-Phyt | SML Phyto Small | SML Phyto Middle | SML Phyto Large | SML CBL Small | SML CBL Middle-Large | SML TEP | SSW Bacteria | SSW HNA | SSW LNA | SSW Pico-Phyt | SSW Phyto Small | SSW Phyto Middle | SSW Phyto Large | SSW CBL Small | SSW CBL Middle-Large | SSW TEP | SSW PPT | SSW Chla |
|---|---|---|---|---|---|---|---|---|---|---|---|---|---|---|---|---|---|---|---|---|---|---|
| Longitude | -.815** | -.842** | -.619** | -.485* | -.812** | -0.462 | 0.292 | 0.209 | -0.034 | 0.249 | -.785** | -.831** | -.516* | -.667** | -.845** | -0.329 | 0.321 | 0.166 | 0.433 | 0.416 | -.630** | -.585** |
| Temperature | 0.427 | 0.377 | 0.265 | 0.191 | 0.184 | 0.255 | .548* | 0.187 | 0.233 | 0.377 | 0.341 | 0.394 | 0.011 | -0.012 | 0.045 | -0.158 | -0.141 | 0.141 | 0.029 | 0.144 | 0.42 | -0.132 |
| Salinity | -.958** | -.927** | -.797** | -0.453 | -.898** | -0.338 | 0.159 | 0.28 | -0.068 | 0.238 | -.957** | -.927** | -.713** | -.565* | -.861** | -0.207 | 0.3 | 0.262 | 0.374 | 0.422 | -0.412 | -0.475 |
| Wind Speed | 0.049 | 0.183 | -0.109 | 0.031 | 0.201 | -0.074 | -.568* | 0.027 | -0.454 | -.578* | 0.054 | 0.114 | -0.054 | 0.232 | 0.275 | 0.079 | -0.047 | 0.187 | 0.087 | 0.078 | .521* | 0.187 |
| T-SML Cd | -0.407 | -0.43 | -0.238 | 0.161 | -0.297 | 0.091 | 0.245 | 0.24 | 0.436 | 0.203 | -0.377 | -0.439 | -0.11 | 0.186 | -0.24 | 0.472 | 0.33 | 0.265 | 0.074 | 0.044 | -0.315 | 0.024 |
| T-SML Co | -0.012 | 0.009 | -0.174 | -0.229 | -0.093 | -0.123 | -0.127 | -0.334 | -0.012 | 0.285 | -0.029 | 0.176 | -0.245 | -0.243 | -0.127 | -0.318 | -.526* | -0.427 | -0.179 | -0.326 | -0.265 | -0.222 |
| T-SML Cu | .654** | .538* | .542* | 0.12 | .520* | 0.069 | -0.165 | -0.118 | -0.066 | -0.182 | .684** | .650** | 0.453 | 0.441 | .507* | 0.026 | -0.305 | -0.077 | 0.066 | -0.468 | 0.3 | 0.191 |
| T-SML Fe | .527* | .515* | .488* | .493* | .505* | .542* | -0.242 | -0.181 | 0.267 | 0.024 | 0.449 | .502* | 0.449 | .493* | 0.456 | 0.33 | -0.15 | -0.141 | -0.373 | -0.421 | 0.424 | 0.371 |
| T-SML Ni | -.610** | -.530* | -.669** | -0.347 | -.539* | -0.223 | -0.15 | 0.063 | -0.181 | 0.268 | -.600* | -0.48 | -.667** | -0.248 | -0.453 | -0.142 | -0.183 | 0.001 | 0.15 | 0.15 | -0.044 | -0.116 |
| T-SML Mo | -0.238 | -0.179 | -0.228 | -0.353 | -0.287 | -0.103 | -0.033 | 0.015 | -0.385 | -0.344 | -0.115 | -0.088 | -0.103 | -0.186 | -0.047 | 0 | -0.234 | -0.053 | -0.118 | -0.191 | -0.206 | -0.288 |
| T-SML Pb | 0.436 | 0.319 | 0.307 | -0.45 | .520* | 0.27 | -0.371 | -0.383 | -0.017 | 0.106 | 0.407 | 0.397 | 0.245 | 0.478 | .593* | 0.43 | -0.17 | -0.444 | -.578* | -0.138 | .524* | .754** |
| T-SML V | -0.096 | -0.039 | -0.201 | 0.34 | -0.137 | -0.328 | -0.114 | -0.244 | -0.358 | -0.229 | -0.066 | 0.135 | -0.257 | -0.387 | -0.076 | -.488* | -0.475 | -0.24 | -0.059 | -0.265 | -0.371 | -0.349 |
| T-SML Zn | .493* | .579* | 0.225 | 0.34 | -.586* | 0.257 | -0.397 | 0.053 | -0.145 | -0.306 | -.561* | .569* | 0.245 | .527* | .627** | 0.166 | -.554* | 0.091 | -0.007 | -0.126 | .594* | 0.371 |
| D-SML Cd | -0.385 | -0.354 | -0.174 | -0.085 | -0.311 | -0.088 | 0.028 | 0.066 | 0.11 | -0.038 | -0.338 | -0.397 | -0.047 | -0.005 | -0.25 | 0.266 | 0.348 | 0.043 | -0.103 | -0.082 | -0.465 | -0.108 |
| D-SML Co | -.583* | -.667** | -0.458 | -0.444 | -.544* | -.490* | 0.369 | -0.069 | 0.037 | 0.412 | -.539* | -.559* | -0.309 | -.627** | -.627** | -0.379 | 0.142 | -0.094 | 0.189 | 0.376 | -.732** | -0.495 |
| D-SML Cu | 0.221 | 0.151 | 0.26 | .712** | 0.272 | .645** | 0.093 | 0.411 | .652** | 0.021 | 0.233 | 0.123 | 0.299 | .738** | 0.343 | .619** | 0.081 | 0.369 | 0.159 | -0.182 | 0.112 | .547* |
| D-SML Fe | -0.301 | -0.327 | -0.078 | 0.08 | -0.115 | -0.069 | -0.163 | 0.039 | 0.083 | 0.221 | -0.346 | -0.471 | -0.044 | -0.047 | -0.223 | -0.056 | 0.404 | 0.082 | 0.279 | 0.259 | -0.029 | 0.279 |
| D-SML Ni | -.926** | -.888** | -.826** | -0.383 | -.825** | -0.276 | 0.253 | 0.267 | 0.009 | 0.356 | -.889** | -.853** | -.690** | -0.462 | -.755** | -0.07 | 0.156 | 0.242 | -0.145 | 0.453 | -0.393 | -0.4 |
| D-SML Mo | -0.056 | -0.042 | -0.034 | 0.311 | -0.074 | 0.431 | .513* | 0.219 | 0.407 | 0.359 | -0.061 | -0.088 | 0.005 | -0.137 | -0.103 | 0.048 | -0.023 | 0.061 | -0.407 | 0.253 | -0.079 | -0.002 |
| D-SML Pb | 0.48 | 0.41 | 0.375 | 0.472 | .512* | .495* | -0.145 | -0.278 | 0.355 | 0.121 | 0.451 | 0.449 | 0.385 | 0.475 | .603* | 0.402 | -0.221 | -0.345 | -0.027 | -0.315 | 0.182 | .732** |
| D-SML V | -0.191 | -0.265 | -0.037 | -.574* | -0.402 | -.569* | 0.146 | -0.361 | -0.282 | 0.174 | -0.277 | -0.208 | -0.152 | -0.468 | -.488* | -0.219 | 0.366 | -0.244 | -0.027 | -0.076 | -0.182 | -.670** |
| D-SML Zn | 0.154 | 0.109 | -0.093 | 0.151 | 0.123 | -0.08 | -0.08 | 0.15 | 0.123 | 0.129 | 0.164 | 0.238 | -0.061 | 0.377 | 0.257 | 0.351 | -0.358 | -0.064 | -0.059 | -0.003 | 0.232 | 0.275 |
| D-SSW Cd | -.826** | -.837** | -.638** | -0.421 | -.809** | -0.285 | 0.333 | 0.48 | -0.206 | 0.068 | -.800** | -.882** | -.579* | -0.429 | -.682** | -0.132 | 0.488 | 0.406 | .512* | 0.443 | -0.211 | -0.28 |
| D-SSW Co | -.586* | -.591* | -.591* | -0.418 | -.608** | -0.439 | 0.35 | -0.025 | 0.049 | 0.491 | -.586* | -.532* | -.537* | -.694** | -.691** | -0.375 | -0.01 | -0.094 | 0.181 | .568* | -.541* | -.538* |
| D-SSW Cu | -0.412 | -0.424 | -0.284 | -0.28 | -0.434 | -0.382 | -0.163 | -0.09 | -0.069 | -0.059 | -0.439 | -0.434 | -0.252 | -0.324 | -0.431 | -0.105 | 0.221 | -0.055 | 0.326 | 0.126 | -0.376 | -0.279 |
| D-SSW Fe | -0.022 | 0.042 | -0.052 | 0.339 | 0.097 | 0.303 | -0.232 | 0.302 | 0.245 | -0.235 | 0.036 | -0.064 | 0.092 | 0.251 | 0.103 | 0.209 | 0.102 | 0.173 | 0.045 | -0.049 | -0.026 | 0.275 |
| D-SSW Ni | -.888** | -.876** | -.721** | -0.422 | -.911** | -0.281 | 0.32 | 0.279 | 0.018 | 0.365 | -.907** | -.899** | -.697** | -.576* | -.874** | -0.151 | 0.365 | 0.2 | 0.304 | 0.462 | -0.389 | -0.497 |
| D-SSW Mo | -0.456 | -0.326 | -0.429 | -0.085 | -0.368 | -0.105 | -0.018 | -0.166 | 0.012 | 0.226 | -.515* | -0.439 | -0.326 | -0.301 | -0.326 | 0.184 | 0.132 | -0.156 | -0.292 | .547* | -0.144 | -0.191 |
| D-SSW Pb | 0.306 | 0.348 | 0.321 | 0.389 | 0.373 | 0.392 | -0.397 | -0.238 | 0.154 | -0.285 | 0.23 | 0.282 | 0.38 | 0.355 | 0.424 | 0.427 | -0.005 | -0.128 | -0.407 | -0.176 | 0.182 | 0.354 |
| D-SSW V | -0.467 | -0.443 | -0.316 | 0.04 | -0.304 | 0.2 | -0.058 | 0.139 | 0.096 | 0.258 | -0.448 | -0.464 | -0.173 | 0.001 | -0.216 | 0.177 | -0.085 | 0.152 | 0.01 | 0.163 | -0.026 | 0.143 |
| D-SSW Zn | -.602* | -.502* | -.534* | -0.113 | -.599* | -0.037 | 0.236 | 0.177 | 0.133 | 0.365 | -.726** | -.591* | -.664** | -0.287 | -.579* | 0.096 | 0.299 | 0.21 | 0.056 | 0.421 | 0.024 | -0.251 |