# Peer review of "Characterising the surface microlayer in the Mediterranean Sea: trace metal concentrations and microbial plankton abundance"

_Biogeosciences, 2019_

## Referee Comment (RC1) · Anonymous Referee #1 · 11 Sep 2019

Review for consideration for its publication in Biogeosciences of Characterising the surface microlayer in the Mediterranean Sea: trace metals concentration and microbial plankton abundance by Antonio Tovar-Sánchez, Araceli Rodríguez-Romero, Anja Engel, Birthe Zäncker, Franck Fu, Emilio Marañón, María Pérez-Lorenzo, Matthieu Bressac, Thibaut Wagener, Karine Desboeuf, Sylvain Triquet, Guillaume Siour, Cécile Guieu.

This manuscript contains the measurement of many ancillary and biological parameters and trace metal concentrations in the surface microlayer and immediate underlaying water collected during a Mediterranean cruise that covered all the main basins of

the western and middle Mediterranean Sea. The manuscript is a fine effort in shedding light in the description of this microenvironment and the parameters that can affect its special biochemical characteristics. Despite its importance for interface processes, not many efforts are dedicated to the surface microlayer and this work is addressed to partially cover this deficit. Due to the amount of work involved and the relevance of the work for the common readers of Biogeosciences I think that the manuscript is well suited for its publication in this journal. The manuscript is well organized although it is obvious that more than one researchers have taken care of different parts, not all of them showing the same skill to write scientific English. Some parts will require grammar revision before publication. I would also miss that they present more data in the text since as it is the reader has to be continuously going back and forth to the tables and those are not reader friendly due to their size. Overall, I would back a major revision decision; the database presented here is very interesting and many parts of the interpretation are very useful but I think that the manuscript can be substantially improved in many aspects.

Before publication, I have three major concerns that the authors need to address: Photorreactions. In a layer so exposed to solar radiation and with a heavy presence of organics prone to form radicals, the authors should have a better understanding of how these processes can affect species distribution in the SML and fluxes off it. However, these reactions are only invoked when the authors cannot explain, with their limited battery of processes, the distribution of a particular trace element. Just as a last resource. And I want to underline that not all metals are equally prone to those effects. It is well known the strong dependence of Cu and Fe redox seawater chemistries on solar radiation. Under strong solar radiation it is very likely that most of Cu and Fe are present as Cu(I) and Fe(II). Then the regular chemistry in seawater shifts, Cu(I) is a weaker acid and binds preferentially weaker acids (S-2) and Fe(II) is far more soluble (6 orders of magnitude!!!) and forms weaker complexes than Fe(III). I have to accept that not much is known about the speciation (organic and redox) of trace elements in the SML but the authors should try to gather all information available and use it for

interpretation. Surprisingly, solar radiation is claimed to play a role in Ni speciation, a metal that is not likely to experience redox changes in seawater conditions (page 13, 15-18). I suggest a better compilation of bibliography referred to photochemical reactions of trace metals in surface waters, clearly identify those metals that can suffer redox reactions and apply this knowledge to the interpretation of distributions from the introduction and not as a last resource. Residence times of trace metals in the SML. There is a section where the authors argue that most of the material in suspension is of Atlantic or European origin except for a few exceptions. Then in order to calculate the residence times of different metals in the SML the authors assume that all metals are present in particles of a certain size except for iron that is in mineral particles ten times higher; and this assumption is for the whole dataset. It is true that if dust is present, its contribution to the rest of the metals measured in this work would be at least 2 orders of magnitude below iron levels (Guieu, Dulac et al. 2010). This supports that Saharan aerosols are not the main source of trace metals. Then why is it suggested that Fe is in thicker particles of "mineral" origin from a different source? Furthermore, there is no relationship between iron levels (high, > 100 ng m-3, in 5 samples) and the proximity to the Sahara or the trajectories shown in the supplementary material or the referred episodes of wet deposition. In my opinion, there is not enough evidence to argue that iron is present in particles of a different nature and those are 10 times bigger. I suggest that the authors repeat calculations assuming all the particles have a common origin and size and then if they want to keep their original assumption, discuss Fe using two scenarios. The use of high regressions as a cause-effect relationship between variables, specifically the whole discussion about Ni toxicity for bacterioplankton. This needs to be toned down several notches. Although possible, high correlations are indicative of a distribution dependent of common causes and not necessarily of a toxic relationship. If that was the case, salinity would be very toxic for bacterioplankton since the regression coefficient is even higher than that of Ni. Ni concentrations in phytoplankton (I am not familiar with bacterioplankton) are quite high (Twining and Baines 2013) despite their limited physiological relevance without causing deleterious effects.

Moreover, in the sampled waters, there is a factor of only two between the highest and the lowest Ni concentrations. It is very unlikely that such a small variation can cause strong toxic effects. I simply do not buy the hypothesis, could be mentioned but only as a hypothesis and I advocate from its removal from abstract and conclusions.

I would also like to see a better explanation about the striking accumulation of certain metals in the SML despite their absence in aerosols (Cd, Mo Pb) even if using bibliographic water column values. I would also like to see d and aerosol mass data in the final version of the manuscript. Comments Page 1 "the total fraction of some reactive metals in the SML (i.e. Cu, Fe, Pb and Zn) showed negative trends with salinity, these trends of concentrations seem to be associate to microbial uptake". Here we have again the problem that a positive or negative high correlation cannot directly be interpreted as a cause-effect relationship. For such statement the authors have first to show that the microbial biomass found in their oligotrophic samples can make a dent in metal concentrations in those waters (from known metal:C ratios). I would find very surprising that the trace element microbial budget is significant when compared to the trace metal phytoplankton budget. Second, why for Cu, Fe, Pb and Zn a negative correlation is indicative of uptake and for Ni is indicative of toxicity? Pb is far more toxic and Ni cellular quotas (at least in phytoplankton, Twining papers) are very high in healthy cells. What are the regression coefficients of those trace elements with respect to salinity? Page 2 5-10 Questions for the authors: Is the relevance of dust deposition also related to the lack of major riverine discharge? The enrichment at surface is not related to the combination of minimum mix with adjacent seas and strong evaporation (close basin)? 13 I would write here may play since most of the following text are considerations and hypotheses. 21 I suggest to define the thickness of this SML or at least what the authors consider here (a brief description of the Wurl formula and the parameters it depends upon) since d data are not shown. 25 The 3 orders of magnitude wide range provided is too much non definition. Are there many different ways to calculate this thickness? Page 3 1 "Characterized by the dominated abundance of microorganisms" bad grammar 3 please remove although. One part of the sentence is not modifying

the other 9 influences 10 "concentrations of Cu, Fe or Pb in the SML increase by a factor of up to 800, 200 and 150 times compared with the underlaying water". Interestingly, this is not the case here. This has to be discussed in detail later on. 18 This is likely long enough to be chemically missing word? and biologically missing word? alter the SML and affect the composition and activity of the neuston community Page 4 Section 2.1 is quite confusing and the quality of English drops substantially. It has to be revised (grammar and spelling) and modified 14 Is this sentence correct and/or complete? It does not make much sense to me. This inlet was developed for sampling both fine and coarse particles, with particles of aerodynamic diameter of about 40 $\mu$m 18 No bibliographic mention to the combination of standard optical and electrical mobility analyzers? 20 a filtration unit 23 all filters / rinsed 23 please rewrite "A sampling strategy was made to avoid the contamination by the cruise smoking" Here add a period and then First 25 the PEGASUS container and the boat's chimney / opposite side of the deck (opposite ship boards?) 28 bad grammar again Page 5 6 Not all metals measured are presented here. Why Cr and Nd are not included? 7 Why rain data are not commented? 18 the glass plate is not conditioned to the seawater matrix before first collection? I wonder how much metal is adsorbed and extracted from the sample from a plate which surface has been activated after acid cleaning and has only be risen with ultrapure water. Can the authors discard that the first extraction of the day is not lower? 21 what was the result of blank checking? Please describe briefly. Here I also warn that if the blank is run immediately after the ultrapure rinsing, metals could be adsorbed by the plate. 23 Wurl's formula? /The total. . .. . . was directly 24 while the. . .. Page 6 5 why only samples for totals were UV digested? Metal organic ligands and DOM were certainly present in the dissolved samples. Cu and Co analysis in dissolved samples are especially dependent in this digestion step (Rapp, Schlosser et al. 2017). 23 Microorganisms in the . . . . . . . . .. . ..were sampled at the same time than. . .. . . ...using a 25 what does it mean "manually sampled"? I hope not what it literally indicates. 29 please split sentence in two. Page 7 Sections 2.3.1 and 2.3.2 are almost free of the bibliographic references where the methodologies have been proved for these specific

purposes. Example: "value of 26,000 $\mu$gC L-1 was used for the concentration of dissolved inorganic carbon", where is this value coming from? 14 I guess fumes were used before filter use. Not clear with the current sequence. 26 linear least squares regression? Page 8 4-5 I find that here the bibliographic revision is too short. There are many more works on the presence of metals in dry aerosols. I would be interested in a very simple study about temporal trends adding studies from the 90s (Roy Chester and several others). In any case the bibliographic search has not been good enough Lines 7-10 Here the discussion is very difficult to follow. Figure 1 does not include sampling dates and figure S1 is confusing with so much overlapping of curves of similar colours. Then it is difficult to follow this discussion. For me it is like all the trajectories do not show Saharan sources but on those two dates the African input was so high that in those cases particle trajectories were "not convenient" and sided for interpretation. Could the authors be clearer about the use of the different information sources? Total mass collected is not provided in the manuscript. 9 loaded with? Lines 15 to 20 In my opinion this section has to be revised by an English native speaker. Furthermore there are comments about data that are not shown in tables or graphs. 27 trace metals conc of. . . . . . . ., with the execption of Pb, were lower than those measured. . . . . . . . . . . in previous MS studies. "In previous studies" but only one manuscript is cited. I stress that the bibliographic search on trace metals in dry deposition in the Mediterranean area has to be extended and results put in that context before publication Page 9 This discussion is very hard to follow unless ranges supporting arguments are provided in the text. It forces the reader to go back and forth to Table 1 that is actually quite hard to read. 7 My question here is how rain affects SML composition and thickness. 8-17 this is a very interesting paragraph. Please discuss the low SML/SSW ratios in the context of the huge ratios referred in the introduction for Cu, Fe and Pb (p 3, 10-11). For Ni, V and Fe the authors should say explicitly that there were no differences between SML and SSW (average close to 1 and standard deviation bigger than the difference). What are the removal processes the authors suggest? Differential dissolution of different metals from the same material? Radiation driven processes? Is taken into account the high efficient mixing in the turbulent 1st meter of the ocean? 18-20. An efficient mixing should be given by close values (as both watermasses mix efficiently they have the same concentrations) and not simply by high regression. If the slope is close to 1, there is good mixing (line constitutes by identical values, if the slope (not r2!!!) is different from 1 that means poor mixing since one of the concentrations is consistently higher than the other and that would mean gradients. Page 10 Cu and Fe experience redox changes as a function of the solar radiation and Pb has a limited solubility of inorganic forms at pH 8. I do not know whether this explains their distribution but I think it is worth mention it. 15-16 this statement disentangling metals from particles sizes is very concerning to me. The statement assumes that 1 Fe is included in some particles and the rest of metals in other particles 2 particles including Fe are so much bigger that sink at 10 times faster speed. I think this requires more discussion, if all metals were part of the same particles and no other process was accounted, this would underestimate Fe residence time by a factor of 10 and its residence time would be perfectly aligned with those of Cu, Zn, V and Pb. First, previous discussion in this manuscript concluded that most of the aerosols had a European or NA origin. Now the authors consider that Fe has a mineral behaviour far from fine anthropogenic particles. Second, I am not familiar with studies showing that fine particles are low in iron with respect to the rest of the meatls in this study, especially those found a t the same order of magnitude. if the rest of the metals come from a different thinner material, and some are at concentrations close to the Fe conc in aerosols, then this thinner material is iron free. Third, this sedimentation velocity through the mixed layer is going to be strongly dependent on the energy of the system and a single value for the whole cruise at any location seems a huge source of error to me. Often we have to make simplistic assumptions but I would like that the authors at least make the effort to discuss the consequences of their decisions in terms of uncertainty. How variable was the mixed layer depth during the cruise? 19 I think the shortest residence time in table 3 is 1.2 minutes and not 12. 24-25 I could not find d values in tables. In Wurl's equation d is a function of the sample volume, number of dips and the screen area with the assumption that the presence of surfactants would

increase the volume retained per dip and therefore d. It is necessary to have d values if we want to evaluate its impact and variability on residence time calculations. Page 11 I would not claim that different behaviours are caused by different reactivities to natural ligands. Of the metals targeted in this study, only Zn has a weak affinity for natural organics (not much is known about V affinity for natural organics). Cu is the clear example of strong affinity to ligands and even is known that this affinity is higher than that for biological membranes (González-Dávila, Santana-Casiano et al. 2000). Here the elephant in the room is photochemical processes. 5 can be known how is d related to wind force?. No consideration of photoreactions? 16 again it is said of other regions but only one example is provided. Rewrite for this specific case or bring more examples. 21 "In general, and with the exception of phytoplankton middle and CBL-small, microbial abundance was higher in the SML than in the SSW with abundances ranging from 1 to 6 times higher for bacteria and CBL-middle-large,respectively (Table 1).". In Table I the groups with a higher concentration in the SML are autotrophs (phyto and cyanobacteria). However, the extremely low Chl-a concentrations in the SSW (low even for oligothropic waters, consistently below 0.1 ug l-1, actually they should revise their numbers, I only saw numbers that low in the eastern mediterranean) point to a lack of viable autotrophs in the SSW. And here it is difficult to point to UV effects since the SML should receive even more radiation. It is a real pain that there are no Chl-a measurements in the SML to infer whether the higher cellular content was constituted by viable cells. It is also shocking the lack of correlation of Chl-a with any of the biological variables. 23 rewrite in English please. Page 12 1-2 It makes sense but I would use could instead of would, it is all speculative. I really doubt that assimilation and storage from such a low biomass could explain trace element trends 5 Revise English. It is very surprising that TEP concentrations (of biological origin) could increase after a dust deposition, they should remain or decrease by scavenging. I would tone down this sentence. First it is based on a single value and second it is not higher than Station 9. 6 "we therefore......" Because there are no correlations between metals and TEP the consequence is metal assimilation by microbes explain longer residence

times? I do not follow the cause-effect relation here. Please include here known Cu, Zn and Fe cellular quotas to justify or discard assimilation (Twining papers). 10 and here appears the elephant in the room. It must be taken into account the complexity of photochemical reactions (reducing Fe and Cu) but also the bleaching effect on DOM and ligands. 11 That Ni is strongly anticorrelated to bacterioplankton is indicative of a relation but not necessarily direct. It could be (as for other metals) that is taken up and it is not toxic; as a possible result the higher the bacterial density, the lower the Ni concentration. Figure 3. Are those least square linear regressions? 17-19 please give data (r2) 23 "close correlated" closely Page 13. There is a lot of discussion about possible mix Atlantic and MS waters but no actual bibliographic search on average values in both waters that could justify that some metals could be enhanced by mixing and others not. Please, look for such data. 16" Indeed, UV radiations in this surface layer are highly intense and can acts as a biochemical microreactor where many transformations and photochemical reaction occurs" rewrite after grammar checking. I find that claiming that photoreactions could explain this bioaccumulation is really far fetched. Specially for a metal that has no different redox states in oxygenated seawater Page 14 "It appears that Nickel-dependent toxicity involving ROS may be likely mechanism of oxidative stress in marine microbial organism of the surface ocean" check grammar but better discard here Conclusions Is Co not affected by chemical and biological processes? That is very surprising due to its important requirement

Figure 1. This is a good figure but I do not understand why has been sent vertical. I guess for the publication will be required a reduction in size, shift to horizontal and increase of the font size. Figure 2. I guess DNi refers to DNi in the SSW. Please reduce size. I am not sure this relationship deserves a whole figure. 1 the regression coefficients are in the tables. Second, the supposed bacterioplankton control by Ni toxicity is a nice hypothesis but data do not prove such dependence. Tables are quite difficult to read and I wonder if these will be legible in the final version of the manuscript. In any case all provide useful information and I would not simply remove data from them. Table 3. Station not satation

Figure S3. Wrong caption.

González-Dávila, M., J. M. Santana-Casiano and L. M. Laglera (2000). "Copper adsorption in diatom cultures." Marine Chemistry 70(1-3): 161-170. Guieu, C., F. Dulac, K. Desboeufs, T. Wagener, E. Pulido-Villena, J.-M. Grisoni, F. Louis, C. Ridame, S. Blain and C. Brunet (2010). "Large clean mesocosms and simulated dust deposition: a new methodology to investigate responses of marine oligotrophic ecosystems to atmospheric inputs." Rapp, I., C. Schlosser, D. Rusiecka, M. Gledhill and E. P. Achterberg (2017). "Automated preconcentration of Fe, Zn, Cu, Ni, Cd, Pb, Co, and Mn in seawater with analysis using high-resolution sector field inductively-coupled plasma mass spectrometry." Analytica Chimica Acta 976: 1-13. Twining, B. S. and S. B. Baines (2013). "The trace metal composition of marine phytoplankton." Annual Review of Marine Science 5: 191-215.

---

## Referee Comment (RC2) · Anonymous Referee #2 · 16 Sep 2019

This manuscript describes trace metal concentrations, along with biological parameters, in aerosols, sea-surface microlayer (SML), and surface waters (SSW) at a number of stations in the Mediterranean Sea. The data appear to be carefully collected and analyzed and of high quality. However, the manuscript is marred by numerous grammatical and other errors, and it needs to be thoroughly edited before it would be acceptable for publication.

However, more substantively, the manuscript doesn't tell much of a story about most of the data, and a few of the relationships that it does present are not supported. In general, it focusses on correlations between parameters, and these can always be

tricky. If a chemical and biological parameter are negatively correlated, is it because the chemical is exerting a toxic influence? Or because biology is drawing down the chemical? Or because both are being differentially affected by something else entirely. We just don't know, but this ms makes some unsupported conclusions nonetheless.

Specific comments There are numerous mis-spellings, grammatical and punctuation errors throughout the manuscript that should be corrected. I have highlighted a subset of these encountered in the first half of the manuscript here, but the entire manuscript needs careful attention and correction of these.

Additionally, I the ms would benefit from more general written description of trends in the results (or state that there are no trends). Currently the reader is left largely to pick their way through the massive, tiny-font tables.

Title: should be "trace metal concentrations" P2, L2: what does it mean for an ecosystem to be "ecologically regulated"? P3, L15: should be "underlying" P4, L15: are the fine or the coarse particles <40 um? P4, L19: year for Rupprecht & Patashnick? P4, L24: what is meant by "the cruise smoking"? P4, L18: how was the sample collected off the glass plate? With a water or acid rinse? If so, how was dilution of original sample estimated? P5, L22: double citation (also check for other instances of this) P6, L4: dissolved samples were not irradiated? Was a particulate CRM analyzed? What was the digestion approach for the particulate/total phase? How long were samples acidified for? P7, L4: I assume this should be 0.5 um P8, L28: here and later the text to refers to "previous studies" but only one study is cited P9, L3: is it standard deviation or error presented here and in other parentheses? P10, L18: in the discussion of residence times the ms refers to particulate metals, but the methods only describes collecting a dissolved (filtered) and a total (unfiltered) fraction. So, how can the behavior of the particulate fraction be isolated and determined? Please explain the assumptions made to do this, so they can be evaluated. P11, L4: what does "dynamic" mean here? P11, L7: this is an example of the selective explanation of elemental behaviors. Is the influence of wind on just Co, or Co and Ni? Additionally, why would wind effect only one or two of

the metals? Would wind not have the same physical transport or diffusion and mixing effects on all metals? Please provide some additional discussion of this very selective effect. P11, L11: it should be "nucleic acid" P11, L13: the methods for characterizing all of these separate biological groups needs to be provided and justified: why were these groups characterized? P11, L24: how is SML fraction different from T-SML? P12, L2: bacterial assimilation would result in no change in T-SML, which includes both dissolved and particulate fractions. P12, L23: why would regeneration in the east only be active for Co, when many other bioactive metals are also remineralized actively? P13: I think there is inadequate support for the conclusion that Ni is inhibiting growth in the surface waters, particularly given the lack of relationship with primary production and chl a. P14, L13: how can there be a 'major difficulty of mobility' (a strange term, I feel) for Fe when the residence times are only a few minutes (table 3)?

Fig S2: please include the year in the date in the caption Fig S3: the figure does not match the caption

---

## Author Comment (AC1) · 24 Oct 2019

We really thank the reviewers for their constructive comments. We believe that this new version of the manuscript has been improved significantly by the reviewers' suggestions as we have addressed all the points they raised. Generally, the text has been modified and we believe it has significantly improved. Specifically, grammatical and syntactic mistakes have been corrected and in general the text has been grammatically reviewed by an English native speaker. We have incorporated further clarification in all section of the manuscript; we have modified tables and figures, incorporated new references and elaborated new figures and tables. The response and actions taken to accommodate

the reviewers' comments are described in the following pages.

Reviewers' Comment: This manuscript contains the measurement of many ancillary and biological parameters and trace metal concentrations in the surface microlayer and immediate underlaying water collected during a Mediterranean cruise that covered all the main basins of the western and middle Mediterranean Sea. The manuscript is a fine effort in shedding light in the description of this microenvironment and the parameters that can affect its special biochemical characteristics. Despite its importance for interface processes, not many efforts are dedicated to the surface microlayer and this work is addressed to partially cover this deficit. Due to the amount of work involved and the relevance of the work for the common readers of Biogeosciences I think that the manuscript is well suited for its publication in this journal. The manuscript is well organized although it is obvious that more than one researchers have taken care of different parts, not all of them showing the same skill to write scientific English. Some parts will require grammar revision before publication. I would also miss that they present more data in the text since as it is the reader has to be continuously going back and forth to the tables and those are not reader friendly due to their size. Overall, I would back a major revision decision; the database presented here is very interesting and many parts of the interpretation are very useful but I think that the manuscript can be substantially improved in many aspects.

Authors' Response: We are very grateful by the deeply review and comments made by this referee, we believe that his/her comments and suggestions have helped to improve significantly the manuscript. The manuscript has been revised grammatically by a native English speaker. We have included more data during the discussion in the different sections of the manuscript.

Reviewers' Comment: Before publication, I have three major concerns that the authors need to address: Photorreactions. In a layer so exposed to solar radiation and with a heavy presence of organics prone to form radicals, the authors should have a better understanding of how these processes can affect species distribution in the SML

and fluxes off it. However, these reactions are only invoked when the authors cannot explain, with their limited battery of processes, the distribution of a particular trace element. Just as a last resource. And I want to underline that not all metals are equally prone to those effects. It is well known the strong dependence of Cu and Fe redox seawater chemistries on solar radiation. Under strong solar radiation it is very likely that most of Cu and Fe are present as Cu(I) and Fe(II). Then the regular chemistry in seawater shifts, Cu(I) is a weaker acid and binds preferentially weaker acids (S-2) and Fe(II) is far more soluble (6 orders of magnitude!!!) and forms weaker complexes than Fe(III). I have to accept that not much is known about the speciation (organic and redox) of trace elements in the SML but the authors should try to gather all information available and use it for interpretation. Surprisingly, solar radiation is claimed to play a role in Ni speciation, a metal that is not likely to experience redox changes in seawater conditions (page 13, 15-18). I suggest a better compilation of bibliography referred to photochemical reactions of trace metals in surface waters, clearly identify those metals that can suffer redox reactions and apply this knowledge to the interpretation of distributions from the introduction and not as a last resource.

Authors' Response: We agree with the reviewer on the importance that the solar radiation has on the redox chemistry of the highly particle reactive elements, such as Fe and Cu. We have now included some discussion on this topic in the manuscript (section 3.2.1). We are also aware that Ni in seawater is thought to occur partly as stable organic complexes and with dissociation rates of its complexes much lower than Fe or Cu (104 times lower in the case of Cu; Morel and Hering, 1993). However, we believe that this dissociation rate could be significantly accelerated by the photochemical reactions and therefore directly affecting its speciation distributions and biological uptake and response. Although the interactions of Ni with dissolved organic matter have not been well studied in seawater, it is thought to occur partly as stable organic complexes and with slow dissociation rates (eg. Jiann et al., 2005; Wen et al. 2011 and reference therein). However, it is known that intense UV radiation can alters concentration, structure, reactivity and metal binding capacity of the organic matter increasing the

proportion of free metals ion and their bioavailability and/or potential toxicity (Cheloni and Slaveykova, 2018). We have included this discussion in the text (section 3.2.4).

- Jiann, K.-T., Wen, L.-S., Santschi, P.H., 2005. Trace metal (Cd, Cu, Ni and Pb) partitioning, affinities and removal in the Danshuei River estuary, a macro-tidal, temporally anoxic estuary in Taiwan. Marine Chemistry 96, 293-313. - Morel, F.M.M., Hering, J.G., 1993. Principles and Applications of Aquatic Chemistry. Wiley, New York, p. 400. - Wen, L.-S., Santschi, P. H., Warnken, K. W., Davison, W., Zhang, H., Li, H.-P. and Jiann, K.-T.: Molecular weight and chemical reactivity of dissolved trace metals (Cd, Cu, Ni) in surface waters from the Mississippi River to Gulf of Mexico, Estuarine, Coastal and Shelf Science, 92(4), 649–658, doi:10.1016/j.ecss.2011.03.009, 2011. - Cheloni, G. and Slaveykova, V.: Combined Effects of Trace Metals and Light on Photosynthetic Microorganisms in Aquatic Environment, Environments, 5(7), 81, doi:10.3390/environments5070081, 2018

Reviewers' Comment: Residence times of trace metals in the SML. There is a section where the authors argue that most of the material in suspension is of Atlantic or European origin except for a few exceptions. Then in order to calculate the residence times of different metals in the SML the authors assume that all metals are present in particles of a certain size except for iron that is in mineral particles ten times higher; and this assumption is for the whole dataset. It is true that if dust is present, its contribution to the rest of the metals measured in this work would be at least 2 orders of magnitude below iron levels (Guieu, Dulac et al. 2010). This supports that Saharan aerosols are not the main source of trace metals. Then why is it suggested that Fe is in thicker particles of "mineral" origin from a different source? Furthermore, there is no relationship between iron levels (high, > 100 ng m-3, in 5 samples) and the proximity to the Sahara or the trajectories shown in the supplementary material or the referred episodes of wet deposition. In my opinion, there is not enough evidence to argue that iron is present in particles of a different nature and those are 10 times bigger. I suggest that the authors repeat calculations assuming all the particles have a common origin

and size and then if they want to keep their original assumption, discuss Fe using two scenarios.

Authors' Response:.The measurements of aerosol composition during the cruise show a positive correlation between Al and Fe atmospheric concentrations whatever the period and with an enrichment factor for Fe close to 1, meaning a main crustal source for Fe (see attached figure R1). This result is consistent with the literature which show that the Fe deposition in Mediterranean Sea is mainly associated to mineral dust particles whatever the period of year, even during the period when air masses are from European region (Guieu et al., 2010, Desboeufs et al.,2018). Even if a part of iron is anthropogenic and associated to fine particles, this fraction is negligible (in mass) in comparison to iron dust-bearing. So, we added these arguments in the text to explain the choice to use a velocity of 1cm/s for Fe. The text now reads: " During the cruise, Al and Fe atmospheric concentrations were correlated at all the stations and the ratio Fe/Al is typical of a crustal source (Fu et al., in prep.). It is known that the atmospheric iron deposition fluxes are associated to mineral dust particles even during the period when the Saharan dust inputs are very low (Desboeufs et al., 2018; Guieu et al., 2010). On the contrary, no correlation with Al is observed for the other metals, except during FAST1-3.". Also, we realized that we made a mistake in the residence time calculation because we used the aerosol metal flux of the first station (station 1) to estimate the residence time of all stations. We have revised and corrected the calculations and now residence time is calculated using the aerosol flux for each station. Recalculated residence times are of the same order of magnitude than before, however it changed the discussion. For example, now we don't have any significant correlation of residence time to wind speed (see table 3). In the previous version residence time of Co was very well correlated with wind speed, which opened the question on the lack of effect of wind speed on the other metals. Now the relative low wind speed during our campaign (9 $\pm$ 4.99 knots) did not affect the residence time of metals in the SML. It has been indicated the section 3.2.2. - Desboeufs, K., Bon Nguyen, E., Chevaillier, S., Triquet, S., and Dulac, F.: Fluxes and sources of nutrient and trace metal

atmospheric deposition in the northwestern Mediterranean, Atmos. Chem. Phys., 18, 14477-14492, https://doi.org/10.5194/acp-18-14477-2018, 2018. - Guieu, C., Loÿe-Pilot, M.-D., Benyahya, L. and Dufour, A.: Spatial variability of atmospheric fluxes of metals (Al, Fe, Cd, Zn and Pb) and phosphorus over the whole Mediterranean from a one-year monitoring experiment: Biogeochemical implications, Marine Chemistry, 120(1–4), 164–178, doi:10.1016/j.marchem.2009.02.004, 2010.

Reviewers' Comment: The use of high regressions as a cause-effect relationship between variables, specifically the whole discussion about Ni toxicity for bacterioplankton. This needs to be toned down several notches. Although possible, high correlations are indicative of a distribution dependent of common causes and not necessarily of a toxic relationship. If that was the case, salinity would be very toxic for bacterioplankton since the regression coefficient is even higher than that of Ni. Ni concentrations in phytoplankton (I am not familiar with bacterioplankton) are quite high (Twining and Baines 2013) despite their limited physiological relevance without causing deleterious effects. Moreover, in the sampled waters, there is a factor of only two between the highest and the lowest Ni concentrations. It is very unlikely that such a small variation can cause strong toxic effects. I simply do not buy the hypothesis, could be mentioned but only as a hypothesis and I advocate from its removal from abstract and conclusions.

Authors' Response: The reviewer is right and we agree that a high correlation between two parameters means a relationship but not necessarily a cause-effect. During the preparation of the manuscript we deeply discussed this point and we concluded that toxicity could be possible although difficult to demonstrate with the available data. Some clues that supported our hypothesis were: 1) the strongly negative correlations between dissolved Ni and microbial abundance; 2) Ni toxicity in the same region was previously suggested although in that case with concentrations 13 times higher than measured during our camping; and 3) although we agree with the reviewer that Ni is not likely to experience redox changes in seawater conditions, the intense UV radiation on the SML can affect the binding capacity of the organic matter and affect to

its bioavailability and/or potential toxicity. Even so, we are aware that this hypothesis remains speculative and we don't have enough information to demonstrate it. As suggested, we have toned down this conclusion along the manuscript. For example, the last sentence of the abstract (i.e. "Our results suggest a toxic effect of Ni on neuston and microbiology community's abundance of the top meter of the surface waters of the Western Mediterranean Sea") has been replaced by "Our results show a strong negative correlation between the Ni concentration and heterotrophic bacterial abundance in the SML and SSW, but we cannot ascertain whether this correlation reflects a toxicity effect or is the result of some other process." We have also modified the last sentences in the conclusion's section, and now reads: "A strong negative correlation between the Ni concentration and heterotrophic bacterial abundance in the SML and SSW could be suggestive of an inhibiting role of this element on the microbial growth in the top metre of the surface; however, further research is needed to confirm this finding."

Reviewers' Comment: I would also like to see a better explanation about the striking accumulation of certain metals in the SML despite their absence in aerosols (Cd, Mo Pb) even if using bibliographic water column values.

Authors' Response: The accumulation of metals in the SML is controlled mainly by their particle-reactive properties. Thus, Cd and Mo are not enriched in the SML while that Pb is, together with Fe and Cu, concentrated in this layer as result of their binding capacity to particles and organic matter. This has been demonstrated in other regions under the influence of very different sources (e.g. ice or African dust) (Tovar-Sánchez et al. 2019). On the other hand, in addition to aerosols there are other sources that can influence the metal composition of the SML, such as floating material (mainly biological) coming from the water column.

-Tovar-Sánchez, A., González-Ortegón, E. and Duarte, C. M.: Trace metal partitioning in the top meter of the ocean, Science of The Total Environment, 652, 907–914, doi:10.1016/j.scitotenv.2018.10.315, 2019.

Reviewers' Comment: I would also like to see d and aerosol mass data in the final version of the manuscript. Authors' Response: We have included these data in a new figure (Figure S3).

Reviewers' Comment: Comments Page 1 "the total fraction of some reactive metals in the SML (i.e. Cu, Fe, Pb and Zn) showed negative trends with salinity, these trends of concentrations seem to be associate to microbial uptake". Here we have again the problem that a positive or negative high correlation cannot directly be interpreted as a cause-effect relationship. For such statement the authors have first to show that the microbial biomass found in their oligotrophic samples can make a dent in metal concentrations in those waters (from known metal:C ratios). I would find very surprising that the trace element microbial budget is significant when compared to the trace metal phytoplankton budget. Second, why for Cu, Fe, Pb and Zn a negative correlation is indicative of uptake and for Ni is indicative of toxicity? Pb is far more toxic and Ni cellular quotas (at least in phytoplankton, Twining papers) are very high in healthy cells.What are the regression coefficients of those trace elements with respect to salinity?

Authors' Response: In this case, we refer only to the total (unfiltered) fraction of the SML, which include the microbial metal pool (the dissolved fraction of the SML were not correlated either with underlayer water or salinity gradient). Again, the reviewer is right in the fact that conclusion is obtained from linear regressions between parameters. However, in this case the variations of metal concentrations in the T-SML are also positive and significantly correlated with the microbial abundance. We are aware that total fraction includes also lithogenic material, however aerosols metal concentrations did not show any longitudinal trend and no other natural or anthropogenic sources were identified in the region, therefore we think that biological uptake could be reasonable cause. We have clarified this and sentence in page 1 reads: "In contrast, the total fraction of some reactive metals in the SML (i.e. Cu, Fe, Pb and Zn) showed a negative correlation with salinity and a positive correlation with microbial abundance, which might be associated with microbial uptake". In the case of Ni, we found the negative

correlation with microbial abundance in both compartments, SML and SSW, and in both fractions, dissolved and total. Therefore, toxicity could be a plausible cause. In the case of Pb, there is not a significant trend with salinity. And the correlations with biological abundance is, when significant, positive.

Reviewers' Comment: Page 2 5-10 Questions for the authors: Is the relevance of dust deposition also related to the lack of major riverine discharge? The enrichment at surface is not related to the combination of minimum mix with adjacent seas and strong evaporation (close basin)?

Authors' Response: Of course. The lack of river discharges makes more significant the contribution of aerosol deposition to the metal budget in the water column of the MS. We agree that, in a mass balance, we should consider the fact that evaporation exceed precipitation and river discharges in the MS, generating a hydric deficit that is partially compensated with a limited water exchange with the Atlantic Ocean through the narrow Strait of Gibraltar. However, although many processes contribute to the enrichment of the surface water in some point (e.g. submarine groundwater discharges), we consider the aerosol deposition as the more significant in the top meter of the water column of the MS.

Reviewers' Comment: 13 I would write here may play since most of the following text are considerations and hypotheses.

Authors' Response: Done. The text now reads: "For example, it has been hypothesized that the high Co concentrations in the MS may stimulate "de novo" synthesis of vitamin B12 as Co is the central metal ion in the B12 molecule (Bonnet et al., 2013)".

Reviewers' Comment: 21 I suggest to define the thickness of this SML or at least what the authors consider here (a brief description of the Wurl formula and the parameters it depends upon) since d data are not shown.

Authors' Response: We have included the Wurl formula and described the parameters

used in the method section 2.2.1.

Reviewers' Comment: 25 The 3 orders of magnitude wide range provided is too much non definition. Are there many different ways to calculate this thickness?

Authors' Response: Generally, the wide range of the thickness of the SML used for different authors is not a calculation issue but rather a sampling issue. The system used for sampling this layer provide you with more or less thickness (e.g. glass plate, rotating drum or screen). In the case of the glass plate sampler used here the thickness is typically 20-150 $\mu$m (Cunliffe and Wurl, 2014). - Cunliffe M., Wurl O. (2014). Guide to Best Practices to Study the Ocean's Surface. Plymouth: Occasional Publications of the Marine Biological Association. https://www.oceanbestpractices.net/handle/11329/261

Reviewers' Comment: Page 3 1 "Characterized by the dominated abundance of microorganisms" bad grammar. Authors' Response: It has been corrected. Now reads: "Characterized by a high abundance of microorganisms...."

Reviewers' Comment: 3 please remove although. One part of the sentence is not modifying the other 9 influences Authors' Response: Corrected.

Reviewers' Comment: 10 "concentrations of Cu, Fe or Pb in the SML increase by a factor of up to 800, 200 and 150 times compared with the underlaying water". Interestingly, this is not the case here. This has to be discussed in detail later on. Authors' Response: Responded below.

Reviewers' Comment: 18 This is likely long enough to be chemically missing word? and biologically missing word? alter the SML and affect the composition and activity of the neuston community Authors' Response: We have changed the sentence. Now reads: "This is likely to be long enough to alter the SML chemically and biologically and affect the composition and activity of the neuston community".

Reviewers' Comment: Page 4 Section 2.1 is quite confusing and the quality of English drops substantially. It has to be revised (grammar and spelling) and modified 14 Is this

sentence correct and/or complete? It does not make much sense to me. This inlet was developed for sampling both fine and coarse particles, with particles of aerodynamic diameter of about 40 $\mu$m Authors' Response: We modified the sentence for clarification and we have revised and corrected the English grammar in the section and along the manuscript. For example, regarding 40 $\mu$m diameter we have indicated that: "This inlet was developed to sample particles with an aerodynamic diameter inferior to 40 $\mu$m (Rajot et al., 2008)."

Reviewers' Comment: 18 No bibliographic mention to the combination of standard optical and electrical mobility analyzers?

Authors' Response: This combination is the association of two commercial instruments to have a large spectrum of sizes: an OPC for particle size distribution from 0.25 to 32 $\mu$m diameter and a SMPS for particle size distribution from 10 nm to 450 nm diameter. We have added the information about these instruments and the text now reads: "The aerosol size distribution from 10 nm to 30 $\mu$m was measured by a combination of standard optical and electrical mobility analysers (SMPS, TSI Scanning Mobility Particle Sizer and GRIMM Inc. optical particle counters - OPC, 1.109)."

Reviewers' Comment: 20 a filtration unit 23 all filters / rinsed 23 please rewrite "A sampling strategy was made to avoid the contamination by the cruise smoking" Here add a period and then First 25 the PEGASUS container and the boat's chimney / opposite side of the deck (opposite ship boards?) 28 bad grammar again. Authors' Response: This sections has been corrected and where necessary rewritten.

Reviewers' Comment: Page 5 6 Not all metals measured are presented here. Why Cr and Nd are not included? Authors' Response: Since these two elements (and also Mn) were not analyzed in surface waters and either discussed for aerosol interpretation, they have been removed.

Reviewers' Comment: 7 Why rain data are not commented? Authors' Response: Concentration of metals in rain has been included in section 3.1.

[Figure]

Reviewers' Comment: 18 the glass plate is not conditioned to the seawater matrix before first collection? I wonder how much metal is adsorbed and extracted from the sample from a plate which surface has been activated after acid cleaning and has only be risen with ultrapure water. Can the authors discard that the first extraction of the day is not lower?

Authors' Response: After rinsed with ultrapure water in the lab on board, the glass plate (and the whole glass plate system) were rinsed in station with seawater several times, and before sample collection the three first dips (SML samples) were discharged. This info has been now indicated in the manuscript and the text in section 2.2.1 now reads: "Once at the station, the glass plate and the whole sampler were rinsed with seawater several times, and the three first dips (SML samples) were discharged".

Reviewers' Comment: 21 what was the result of blank checking? Please describe briefly. Here I also warn that if the blank is run immediately after the ultrapure rinsing, metals could be adsorbed by the plate.

Authors' Response: In all cases blank signals were always lower than 20% of the sample signals for all elements. This has been indicated in the text that now reads: "The sample signal to blank ratio was typically greater than 5:1 for all elements". We really don't know how much metals from ultrapure water are adsorbed to the plate but we assume that it will be equal or less than those absorbed from seawater samples.

Reviewers' Comment: 23 Wurl's formula? /The total. . .. . . was directly 24 while the. . .. Authors' Response: Wurl's formula has been included. And grammatical errors corrected.

Reviewers' Comment: Page 6 5 why only samples for totals were UV digested? Metal organic ligands and DOM were certainly present in the dissolved samples. Cu and Co analysis in dissolved samples are especially dependent in this digestion step (Rapp, Schlosser et al. 2017).

[Figure]

Authors' Response: The reviewer is right in the fact that organic ligand and DOM are present in the dissolved fraction. In the case of Rapp et al 2017, the UV digestion has been demonstrated to be necessary for a complete or better determination of Co and Cu using the SeaFAST (i.e. using a particulate chelating resin). In our case, where an organic liquid-liquid extraction using APDC-DDDC was used, the UV radiation is not considered a critical step. However, in the case of the total fraction, and awarded of the high content of MO, we decided to include the UV step to guarantee a full cell breakdown and the complete digestion.

Reviewers' Comment: 23 Microorganisms in the . . .. . .. . .. . ..were sampled at the same time than. . .. . ...using a Authors' Response: Corrected

Reviewers' Comment: 25 what does it mean "manually sampled"? I hope not what it literally indicates. Authors' Response: For sampling the seston in the SSW the bottle was dipping directly in the water. To do that and to avoid any organic contamination, hand and arm, were covered with gloves and sleeve protectors.

Reviewers' Comment: 29 please split sentence in two Authors' Response: Done

Reviewers' Comment: Page 7 Sections 2.3.1 and 2.3.2 are almost free of the bibliographic references where the methodologies have been proved for these specific purposes. Example: "value of 26,000 $\mu$gC L-1 was used for the concentration of dissolved inorganic carbon", where is this value coming from?

Authors' Response: We have included more bibliographic references to support our statements. Primary production rates were calculated by taking into account the mean concentration of dissolved inorganic carbon (DIC) measured during the cruise (26,661 ug/L). The relevant sentences in section 2.3.2 have been modified accordingly.

References included in this section are: - Harvey G. Microlayer collection form the sea surface> a new method and initial results. Limnol Ocean 1966; 11: 608-613. - Cunliffe M, Wurl O. Guide to Best Practices to Study the Ocean's Surface. Occas Publ Mar Biol

Assoc United Kingdom 2014. - Marie D, Partensky F, Jacquet S, Vaulot D. Enumeration and Cell Cycle Analysis of Natural Populations of Marine Picoplankton by Flow Cytometry Using the Nucleic Acid Stain SYBR Green I. Applied and Environmental Microbiology 1997; 63:186-193.

Reviewers' Comment: 14 I guess fumes were used before filter use. Not clear with the current sequence. Authors' Response: After the filtration, filters were exposed to HCl fumes in order to remove the non-fixed, inorganic 14C. The sentence has been clarified accordingly.

Reviewers' Comment: 26 linear least squares regression? Authors' Response: Yes, corrected.

Reviewers' Comment: Page 8 4-5 I find that here the bibliographic revision is too short. There are many more works on the presence of metals in dry aerosols. I would be interested in a very simple study about temporal trends adding studies from the 90s (Roy Chester and several others). In any case the bibliographic search has not been good enough.

Authors' Response: We have included more references in this section. We have added a sentence about the temporal trend to support our observation. The text reads: "However, it has been shown that aerosol concentrations of anthropogenic trace metals (i.e. Pb, Cd and Zn) have decreased remarkably over the last two decades, while crustal metals have not shown any evolution (Heimbürger et al., 2010)."

Reviewers' Comment: Lines 7-10 Here the discussion is very difficult to follow. Authors' Response: It has been rewritten.

Reviewers' Comment: Figure 1 does not include sampling dates and figure S1 is confusing with so much overlapping of curves of similar colours. Then it is difficult to follow this discussion. For me it is like all the trajectories do not show Saharan sources but on those two dates the African input was so high that in those cases particle trajectories

were "not convenient" and sided for interpretation. Could the authors be clearer about the use of the different information sources? Total mass collected is not provided in the manuscript.

Authors' Response: Date and time of sampling stations are given in Table 1. We think that including dates on the Figure 1 will make the figure less legible. We have indicated the period of sampling in the figure caption. We have included the station name in the Figure S1 for better interpretation. Also, attached is the backward trajectories for the stations 9 and Fast1-4 (Figure R2), and only stations 9 and Fast 1-2 could be influence from African dust. It has been modified in the manuscript that now reads: "The composition of metal aerosols was mainly influenced by air masses from the North of Europe and Atlantic Ocean (Figure S1), except between June 1st and June 5th (i.e. for the stations St 9 and Fast 1-4) when African air masses were loaded with dust (Figure S1-2)." Total aerosol mass collected is now provided in a new figure (Figure S3). Reviewers' Comment: 9 loaded with? Authors' Response: Yes, corrected.

Reviewers' Comment: Lines 15 to 20 In my opinion this section has to be revised by an English native speaker. Furthermore there are comments about data that are not shown in tables or graphs.

Authors' Response: We have revised and corrected the English grammar in this section and along the manuscript. We have included new figure that refers to cited data.

Reviewers' Comment: 27 trace metals conc of. . .. . ..., with the execption of Pb, were lower than those measured. . .. . .. . .. . . in previous MS studies. "In previous studies" but only one manuscript is cited. I stress that the bibliographic search on trace metals in dry deposition in the Mediterranean area has to be extended and results put in that context before publication

Authors' Response: Unfortunately, up to our knowledge the only data of trace metals in the SML in the MS is from Tovar-Sánchez et al., 2014. Nevertheless, as indicated by the reviewer we have extended the biographic on trace metals in dry deposition in

the MS.

Reviewers' Comment: Page 9 This discussion is very hard to follow unless ranges supporting arguments are provided in the text. It forces the reader to go back and forth to Table 1 that is actually quite hard to read. Authors' Response: To facilitate the reading we have included ranges to support our statements.

Reviewers' Comment: 7 My question here is how rain affects SML composition and thickness Authors' Response: The atmospheric fluxes of trace metals in the dust wet deposition event are higher that the dry deposition fluxes estimated from aerosol concentrations during the dusty period (Fu et al., in prep). The wet deposition enables to wash-out all the thickness of atmospheric boundary layer and in particular dust which are transported in altitude (Desboeufs et al., 2010). Moreover, the higher concentration of metals in rain depositions is due to the light rain event and low volumes of rain collected during our campaign as consequence of the so-called 'wash-out' effect at the onset of rain events (Helmers and Schrems, 1995; Chance et al. 2015). This could explain why during Fast 3 (affected by the dusty rain events) the concentration of some metals in the T-SML were significantly high. We think that the slight rain event did not affect the thickness of the SML. We have included the total trace metals concentrations in the dusty rain to support the potential effect of rain on the SML composition during FAST-3 (P10, L8): " The total trace metal concentrations in the dusty rain collected, ranged from 180 pM for Cd to 343 nM for Fe (Cd: 180 pM , Co: 1380 pM , Cu: 18.1 nM, Fe: 343 nM, Ni: 9.9 nM, Mo: 875 pM, V: 26.9 nM, Zn: 345 nM and Pb: 788 pM)."

- Desboeufs K. , E. Journet, J.-L. Rajot, S. Chevaillier, S. Triquet, P. Formenti, and A. Zakou Chemistry of rain events in West Africa: evidence of dust and biogenic influence in convective systems, Atmos. Chem. Phys., 10, 9283-9293, doi:10.5194/acp-10-9283-2010, 2010. - E. Helmers, O. Schrems. Wet deposition of metals to the tropical North and the South Atlantic Ocean Atmos. Environ., 29 (18) (1995), pp. 2474-2484 - R. Chance, et al.Atmospheric trace metal concentrations, solubility and deposition fluxes in remote marine air over the south-east Atlantic Mar. Chem., 177 (Part 1)

[Figure]

(2015), pp. 45-56

Reviewers' Comment: 8-17 this is a very interesting paragraph. Please discuss the low SML/SSW ratios in the context of the huge ratios referred in the introduction for Cu, Fe and Pb (p 3, 10-11). For Ni, V and Fe the authors should say explicitly that there were no differences between SML and SSW (average close to 1 and standard deviation bigger than the difference). What are the removal processes the authors suggest? Differential dissolution of different metals from the same material? Radiation driven processes? Is taken into account the high efficient mixing in the turbulent 1st meter of the ocean?

Authors' Response: The comparison indicated in the introduction is between the unfiltered fraction of the SML versus the dissolved fraction of the underlayer water. This has been clarified in the introduction section, and now reads: "For example, in regions under the influence of dust events, such as the North Atlantic Ocean or Mediterranean Sea, concentrations of Cu, Fe or Pb in the total pool of the SML are up to 800, 200 and 150 times higher than in the dissolved metal pool of the underlying water (Tovar-Sánchez et al., 2019)". The reviewer is right in the fact that for dissolved V and Fe, the differences of concentration between compartments are not significant when averages are compared, however there were significant differences for some particular stations. For example, for Fe (St6): 13.7 nM (D-SML) vs 2.3 nM (SSW); V (ION-1): 38.2 nM (D-SML) vs 18.7 (SSW). We have modified the text and now reads: "The SML to SSW concentration ratio for V ($1.2 \pm 0.42$) and Fe ($1.3 \pm 1.5$) indicated only slight enrichment in the SML over the underlying water, while the ratio for Mo ($1.0 \pm 0.1$) indicated no difference between layers (Table 1)". Since comparation here is between dissolved fractions, diffusion is likely the main mechanism that provide differences among compartments. Although the mechanisms of dissolution processes for each metal in the SML are not been addressed yet, we believe that photoreaction due to intense UV radiation is the main driver processes. This has been now discussed in the manuscript (section 3.2.1.).

Reviewers' Comment: 18-20. An efficient mixing should be given by close values (as both watermasses mix efficiently they have the same concentrations) and not simply by high regression. If the slope is close to 1, there is good mixing (line constitutes by identical values, if the slope (not r2!!!) is different from 1 that means poor mixing since one of the concentrations is consistently higher than the other and that would mean gradients.

Authors' Response: The reviewer is right. We have replaced the word "mixing" by "transfer", the text now reads: ". . .indicating an efficient diffusive transfer between these two compartments for these elements." Now, a high regression is indicative of a good transfer rather a good mixing.

Reviewers' Comment: Page 10 Cu and Fe experience redox changes as a function of the solar radiation and Pb has a limited solubility of inorganic forms at pH 8. I do not know whether this explains their distribution but I think it is worth mention it.

Authors' Response: We have included information on different chemical properties of different metals to highlight the complexity in understanding the solubility process that is happening in SML. We have included in the last paragraph of section 3.2.1. the next information: "On the other hand, the complex matrix of the SML and the particular organic and inorganic speciation of each metal studied in the SML will affect their distribution. Thus, for example, Cd and Zn characterized by an oxidation state number of II can vary from very weak to very strong complexation. Lead in oxygenated seawater is partitioned between chloride and carbonate complexes, whiles Fe and Cu speciation are strongly influenced by pH (Byrne, 2002)."

Reviewers' Comment: 15-16 this statement disentangling metals from particles sizes is very concerning to me. The statement assumes that 1 Fe is included in some particles and the rest of metals in other particles 2 particles including Fe are so much bigger that sink at 10 times faster speed. I think this requires more discussion, if all metals were part of the same particles and no other process was accounted, this would

underestimate Fe residence time by a factor of 10 and its residence time would be perfectly aligned with those of Cu, Zn, V and Pb. First, previous discussion in this manuscript concluded that most of the aerosols had a European or NA origin. Now the authors consider that Fe has a mineral behaviour far from fine anthropogenic particles. Second, I am not familiar with studies showing that fine particles are low in iron with respect to the rest of the meatls in this study, especially those found a t the same order of magnitude. if the rest of the metals come from a different thinner material, and some are at concentrations close to the Fe conc in aerosols, then this thinner material is iron free. Third, this sedimentation velocity through the mixed layer is going to be strongly dependent on the energy of the system and a single value for the whole cruise at any location seems a huge source of error to me. Often we have to make simplistic assumptions but I would like that the authors at least make the effort to discuss the consequences of their decisions in terms of uncertainty. How variable was the mixed layer depth during the cruise?

Authors' Response: The measurements of aerosol composition during the cruise show a positive correlation between Al and Fe atmospheric concentrations whatever the period and with an enrichment factor for Fe close to 1, meaning a main crustal source for Fe. This result is consistent with the literature which show that the Fe deposition in Mediterranean Sea is mainly associated to mineral dust particles whatever the period of year, even during the period when air masses are from European region (Guieu et al., 2010, Desboeufs et al.,2018). Even if a part of iron is anthropogenic and associated to fine particles, this fraction is negligible (in mass) in comparison to iron dust-bearing. So, we added these arguments in the text to explain the choice to use a velocity of 1cm/s for Fe. The text now reads: " During the cruise, Al and Fe atmospheric concentrations were correlated at all the stations and the ratio Fe/Al is typical of a crustal source (Fu et al., in prep.). It is known that the atmospheric iron deposition fluxes are associated to mineral dust particles even during the period when the Saharan dust inputs are very low (Desboeufs et al., 2018; Guieu et al., 2010). On the contrary, no correlation with Al is observed for the other metals, except during FAST1-3.".

- Desboeufs, K., Bon Nguyen, E., Chevaillier, S., Triquet, S., and Dulac, F.: Fluxes and sources of nutrient and trace metal atmospheric deposition in the northwestern Mediterranean, Atmos. Chem. Phys., 18, 14477-14492, https://doi.org/10.5194/acp-18-14477-2018, 2018. - Guieu, C., Loÿe-Pilot, M.-D., Benyahya, L. and Dufour, A.: Spatial variability of atmospheric fluxes of metals (Al, Fe, Cd, Zn and Pb) and phosphorus over the whole Mediterranean from a one-year monitoring experiment: Biogeochemical implications, Marine Chemistry, 120(1–4), 164–178, doi:10.1016/j.marchem.2009.02.004, 2010.

Reviewers' Comment: 19 I think the shortest residence time in table 3 is 1.2 minutes and not 12. Authors' Response: The reviewer was right. However, residence time has been recalculated as explained before.

Reviewers' Comment: 24-25 I could not find d values in tables. In Wurl's equation d is a function of the sample volume, number of dips and the screen area with the assumption that the presence of surfactants would increase the volume retained per dip and therefore d. It is necessary to have d values if we want to evaluate its impact and variability on residence time calculations. Authors' Response: d values and thickness of the SML have been included in Table 4.

Reviewers' Comment: Page 11 I would not claim that different behaviours are caused by different reactivities to natural ligands. Of the metals targeted in this study, only Zn has a weak affinity for natural organics (not much is known about V affinity for natural organics). Cu is the clear example of strong affinity to ligands and even is known that this affinity is higher than that for biological membranes (González-Dávila, Santana-Casiano et al. 2000). Here the elephant in the room is photochemical processes.

Authors' Response: Unfortunately, we don't know yet what process is conditioning the behavior and distribution of each metal in the SML. By sure, it is not only one but the combination of many (e.g. solar radiation, wind speed, OM contents, neuston composition, etc.) that affect different to each metal according to its reactivity and redox

potentials. In this sense we think that the reviewer comments have significantly helped to extend this discussion. We have extended in the manuscript the potential role of photochemistry in the SML.

Reviewers' Comment: 5 can be known how is d related to wind force?. No consideration of photoreactions?

Authors' Response: Yes, wind speed is directly related with the thickness of the SML that in turn it affects the number of dips needed to collect the SML sample. For that reason, wind speed was included as one parameter in the statistical test (Table 4). Since not significant correlation was found between metals concentration and wind forces we have removed the discussion about the effect of wind speed and TSML Co concentration. The text now reads: "Wind speed seems not to have affected the residence time of any metal in the SML (Table 3), which is probably due to the low speed registered during our campaign (9 $\pm$ 4.99 knots) (Table 1).

Reviewers' Comment: 16 again it is said of other regions but only one example is provided. Rewrite for this specific case or bring more examples.

Authors' Response: We have included more references to support this statement. Additional references included are: - Engel A, Galgani L. The organic sea-surface microlayer in the upwelling region off the coast of Peru and potential implications for air-sea exchange processes. Biogeoscienes 2016; 13: 989-1007. - Agogue H, Casamayor E.O., Bourrain M, Obernostererr I, Joux F, Herndl G.J., Lebaron P. A survey on bacteria inhabiting the sea surface microlayer of coastal ecosystems. FEMS Microbiol Ecol 2005; 54: 269-280. - Joux F, Agogue H, Obernosterer I, Dupuy C, Reinthaler T, Herndl G.J., Lebaron P. Microbial community structure in the sea surface microlayer at two contrasting sites in the northwestern Mediterranean Sea. Aquat Microb Ecol 2006; 42: 91-104.

Reviewers' Comment: 21 "In general, and with the exception of phytoplankton middle and CBLsmall, microbial abundance was higher in the SML than in the SSW with abundances ranging from 1 to 6 times higher for bacteria and CBL-middle-large,respectively (Table 1).". In Table I the groups with a higher concentration in the SML are autotrophs (phyto and cyanobacteria). However, the extremely low Chl-a concentrations in the SSW (low even for oligothropic waters, consistently below 0.1 ug l-1, actually they should revise their numbers, I only saw numbers that low in the eastern mediterranean) point to a lack of viable autotrophs in the SSW. And here it is difficult to point to UV effects since the SML should receive even more radiation. It is a real pain that there are no Chl-a measurements in the SML to infer whether the higher cellular content was constituted by viable cells. It is also shocking the lack of correlation of Chl-a with any of the biological variables.

Authors' Response: The low Chl-a values were not anomalous. For comparison one can look at the Prosope (summer/fall) and Boum (summer) cruises: in both cases during most of the longitudinal transect (including the central Med Sea and a good section of the Western Med Sea) surface Chl-a values were below 0.1 ug/L (Crombet et al. 2011 Biogeosciences, 8, 459–475). In fact, the Chl-a concentrations were actually very typical of expected ones from satellite observations climatology. The period of the cruise was chosen to be a compromise between very low Chl-a concentration and high probability of dust deposition. According to Chl-a concentration over the whole Mediterranean Sea, our expedition encountered a classical situation regarding Chl-a (i.e. Bosc et al., 2004) (Figure R3). (this is fully developed in the introduction/strategy paper by Guieu et al. 2019). It is much lower in the eastern Mediterranean that can be qualified as ultraoligotrophic with concentrations < 0.03 $\mu$g.l-1. The suggestion that there were 'no viable autotrophs' in surface waters does not seem correct, because the microcosm dust addition experiments (conducted with water from ca. 5 m) showed a response of the phytoplankton community to the nutrients released from the dust. This response was particularly noticeable in the ION and FAST long-duration stations, but existed also in the G treatment at TYR. We have focused the discussion in the most representative microbial community in the SML (i.e. Bacteria; High nucleic acid-content bacteria: HNA; Low nucleic acid-content bacteria: LNA; pico-phytoplankton). We have

removed the different phytoplankton groups from the text since discussion about their abundance is speculative with the existing data. Also, we have included a paragraph explaining the microbial abundance differences between SSW and SML. The paragraph reads: "Bacterial abundances did not differ significantly between SML and SSW (Table 2). The only slight bacterial enrichment was found after dust input due to an increase in the bacterial cells in the SML, which quickly reverted to the abundances found before the dust input within 48 hours. Phytoplankton was only slightly, but significantly (t-test, p=0.002, n=12) enriched in the SML with an average enrichment of 1.5 compared to the SSW."

- Crombet Y., K. Leblanc, B. Que ÌĄguiner, T. Moutin, P. Rimmelin, J. Ras, H. Claustre, N. Leblond, L. Oriol, and M. Pujo-Pay. Deep silicon maxima in the stratified oligotrophic Mediterranean Sea. Biogeosciences, 8, 459–475, 2011. - Bosc, E., Bricaud, A., & Antoine, D. (2004) Seasonal and interannual variability in algal biomass and primary production in the Mediterranean Sea, as derived from 4 years of SeaWiFS observations, Global Biogeochemical Cycles, 18, GB1005, doi:10.1029/2003GB002034. - Guieu C., D'Ortenzio F., Dulac F., Taillandier V., Doglioli A., Petrenko A., Barrillon S., Mallet M., Nabat P., Desboeufs K., Process studies at the air-sea interface after atmospheric deposition in the Mediterranean Sea: objectives and strategy of the PEACETIME oceanographic campaign (May-June 2017), in prep, this issue, 2019

Reviewers' Comment: 23 rewrite in English please Authors' Response: done.

Reviewers' Comment: Page 12 1-2 It makes sense but I would use could instead of would, it is all speculative. I really doubt that assimilation and storage from such a low biomass could explain trace element trends

Authors' Response: We have toned down this statement. Now it reads: "Bacteria could efficiently assimilate the fraction of Cu, Fe and Zn available, favouring a decrease in the D-SML fraction (Table 1-2)".

Reviewers' Comment: 5 Revise English. It is very surprising that TEP concentrations

(of biological origin) could increase after a dust deposition, they should remain or decrease by scavenging. I would tone down this sentence. First it is based on a single value and second it is not higher tan Station 9 Authors' Response: The reviewer is right. We have removed that sentence.

Reviewers' Comment: 6 "we therefore. . .. . .." Because there are no correlations between metals and TEP the consequence is metal assimilation by microbes explain longer residence times? I do not follow the cause-effect relation here. Please include here known Cu, Zn and Fe cellular quotas to justify or discard assimilation (Twining papers).

Authors' Response: We agree with the reviewer. Since the metal assimilation by microbes could be feasible but we cannot demonstrate it, we have modified the sentence as follow: "Metal assimilation by microbial communities could explain the higher residence time of Cu and Zn (in the order of hours) in the SML, although information about the metal content in seston would be necessary to corroborate this hypothesis."

Reviewers' Comment: 10 and here appears the elephant in the room. It must be taken into account the complexity of photochemical reactions (reducing Fe and Cu) but also the bleaching effect on DOM and ligands. Authors' Response: Yes, we agree. Discussion about this issue has been included in section 3.2.1

Reviewers' Comment: 11 That Ni is strongly anticorrelated to bacterioplankton is indicative of a relation but not necessarily direct. It could be (as for other metals) that is taken up and it is not toxic; as a possible result the higher the bacterial density, the lower the Ni concentration. Authors' Response: We tried to argument this in previous comments. In any case, we have toned down this hypothesis along the ms.

Reviewers' Comment: Figure 3. Are those least square linear regressions? Authors' Response: Yes, they are. It has been indicated in the caption of the Figure 3.

Reviewers' Comment: 17-19 please give data (r2) 23 "close correlated" closely. Authors' Response: Done

Reviewers' Comment: Page 13. There is a lot of discussion about posible mix Atlantic and MS waters but no actual bibliographic search on average values in both waters that could justify that some metals could be enhanced by mixing and others not. Please, look for such data.

Authors' Response: The surface distribution of metals in the western Mediterranean Sea is known to be impacted by the Atlantic inflow water (e.g. Morley et al. 1997; Gómez 2003 and references therein). However, this discussion has been always focused on the surface layer considering the layer below 10 m. Although, undoubtedly it is considered relevant in the global surface distribution of metals, we believe that in the study of the SML other factors such as aerosol deposition or chemical and biochemical issues are more relevant.

- Morley N.H., Burton J.D., Tankere S.P.C. and Martin J-M. 1997. Distribution and behaviour of some dissolved trace metals in the western Mediterranean Sea. Deep-Sea Research II, Vol. 44, No. 34, pp. 675-691. - Gómez F. 2003. The role of the exchanges through the Strait of Gibraltar on the budget of elements in the Western Mediterranean Sea: consequences of human-induced modifications. Marine Pollution Bulletin 46 (2003) 685–694

Reviewers' Comment: 16" Indeed, UV radiations in this surface layer are highly intense and can acts as a biochemical microreactor where many transformations and photochemical reaction occurs" rewrite after grammar checking. I find that claiming that photoreactions could explain this bioaccumulation is really far fetched. Specially for a metal that has no different redox states in oxygenated seawater

Authors' Response: Considering this and previous reviewer's comments we have rewritten this part. Now, we discuss (as follow) the potential role of UV radiation on the dissolution (and bioavailability) of Ni: "The toxicity to phytoplankton of divalent, cationic trace metals, such as Ni or Cu, is probably controlled by its free metal ion

concentration (Donat et al., 1994). Although the Ni interactions with dissolved organic matter have not been studied well in seawater, they are thought to occur partly as stable organic complexes and with slow dissociation rates (Wen et al., 2011). However, intense UV radiation can alter the concentration, structure, reactivity and metal binding capacity of the organic matter, thus increasing the proportion of free metal ions and their bioavailability and/or potential toxicity (Cheloni and Slaveykova, 2018)."

Reviewers' Comment: Page 14 "It appears that Nickel-dependent toxicity involving ROS may be likely mechanism of oxidative stress in marine microbial organism of the surface ocean" check gramar but better discard here

Authors' Response: We have toned down this statement and delete the last sentences. The paragraph finish as follow: "It appears that nickel-dependent toxicity involving ROS could be a mechanism of oxidative stress in microbial organisms of the surface of oceans. While the effect of Ni on microalgae has been studied with laboratory cultures (Brix et al., 2017; Macomber and Hausinger, 2011, 2016), its potential toxic role in the surface of oceans has not yet been investigated."

Reviewers' Comment: Conclusions Is Co not affected by chemical and biological processes? That is very surprising due to its important requirement Authors' Response: We have modified this sentence. Now reads: "While some metals entering the SML (e.g. Cd, Co, Ni and V) show efficient diffusive mixing from the SML to the SSW, more reactive metals such as Cu, Fe, Pb and Zn seem to exhibit a slower diffusion".

Reviewers' Comment: Figure 1. This is a good figure but I do not understand why has been sent vertical. I guess for the publication will be required a reduction in size, shift to horizontal and increase of the font size. Authors' Response: We have changed the format and now is in horizontal.

Reviewers' Comment: Figure 2. I guess DNi refers to DNi in the SSW. Please reduce size. I am not sure this relationship deserves a whole figure. 1 the regression coefficients are in the tables. Second, the supposed bacterioplankton control by Ni toxicity

is a nice hypothesis but data do not prove such dependence.

Authors' Response: DNi refers to both, SML and SSW: now, it is clarified in the caption. We think that this significant correlation disserves to be plotted since it is found in both, SML and SSW. Although we agree with the reviewer that the correlation does not prove the Ni toxicity it is the base for our argumentations.

Reviewers' Comment: Tables are quite difficult to read and I wonder if these will be legible in the final version of the manuscript. In any case all provide useful information and I would not simply remove data from them. Authors' Response: We have splitted the Table 1 in two for easier reading.

Reviewers' Comment: Table 3. Station not satation Authors' Response: Done

Reviewers' Comment: Figure S3. Wrong caption.

Authors' Response: The reviewer is right. Figure S3 (now Figure S4) has been changed with rainfall rates from the radar European composite product that are geo-referenced allowing to plot the position of the FAST station. The figure cation has been changed consequently and now reads: "Accumulated rainfall during the night between June 3rd and 4th 2017 (00h00 – UTC) and position of R/V at the Fast Station. The rainfall rates are estimated from the radar European composite products provided by the Odyssey system." We really thank to the reviewer for all these comments and the many errors detected.

Please also note the supplement to this comment:
https://www.biogeosciences-discuss.net/bg-2019-290/bg-2019-290-AC1-supplement.pdf
* * *
**Figure R1**. Al and Fe atmospheric mass concentrations in all stations along Pacetime cruise (Fu et al. in preparation).

[Figure]

**Fig. 1.**

**Figure R2**. backward trajectories for the stations 9 and Fast1-4

[Figure]

**Fig. 2.**

**Figure R3**. Monthly averaged chlorophyll maps derived from SeaWiFS data for the year 1999. Source Bosc. et al. 2004

[Figure]

**Fig. 3.**

---

## Author Comment (AC2) · 24 Oct 2019

We really thank the reviewers for their constructive comments. We believe that this new version of the manuscript has been improved significantly by the reviewers' suggestions as we have addressed all the points they raised. Generally, the text has been modified and we believe it has significantly improved. Specifically, grammatical and syntactic mistakes have been corrected and in general the text has been grammatically reviewed by an English native speaker. We have incorporated further clarification in all section of the manuscript; we have modified tables and figures, incorporated new references and elaborated new figures and tables. The response and actions taken to accommodate

the reviewers' comments are described in the following pages.

Reviewers' Comment: This manuscript describes trace metal concentrations, along with biological parameters, in aerosols, sea-surface microlayer (SML), and surface waters (SSW) at a number of stations in the Mediterranean Sea. The data appear to be carefully collected and analyzed and of high quality. However, the manuscript is marred by numerous grammatical and other errors, and it needs to be thoroughly edited before it would be acceptable for publication. Authors' Response: We are very grateful by the review and comments made by this referee, we believe that his/her comments and suggestion have helped to improve significantly the manuscript. The manuscript has been revised grammatically by a native English speaker.

Reviewers' Comment: However, more substantively, the manuscript doesn't tell much of a story about most of the data, and a few of the relationships that it does present are not supported. In general, it focusses on correlations between parameters, and these can always be tricky. If a chemical and biological parameter are negatively correlated, is it because the chemical is exerting a toxic influence? Or because biology is drawing down the chemical? Or because both are being differentially affected by something else entirely. We just don't know, but this ms makes some unsupported conclusions nonetheless.

Authors' Response: The reviewer is right. This criticism has been made for both reviewers and we agree with them. We agree that a high regression between two parameters means a relationship but not necessarily a cause-effect. For that reason, we have toned down many of our conclusions and extended our discussion to support them and, at the same time, avoiding being too speculative. During the preparation of the manuscript we deeply discussed the interpretations of these correlations, because, as pointed out by the reviewer, a same linearity could be interpreted either, as toxicity or biological uptake. Although difficult to demonstrate with the existing data, we consider possible the toxicity of Ni by the following reasons: 1) the strongly and negative correlations between dissolved Ni (i.e. bioavailable) and microbial abundance in both,

SML and SSW. This kind of correlation was not found with any other metal; 2) previous study demonstrated Ni toxicity in the same region although with concentrations 13 times higher that our measured background; and 3) we have included now the potential effect that intense UV radiation happening in the SML could have on the binding capacity of colloidal-Ni, and consequently on its bioavailability and/or potential toxicity. Even so, we are aware that this hypothesis remains speculative and we don't have enough information to demonstrate it. Therefore, we have toned down the conclusion along the manuscript.

Reviewers' Comment: Specific comments There are numerous mis-spellings, grammatical and punctuation errors throughout the manuscript that should be corrected. I have highlighted a subset of these encountered in the first half of the manuscript here, but the entire manuscript needs careful attention and correction of these.

Authors' Response: We thanks to the reviewer for his/her grammar revision. We have corrected all of them and revised the entire manuscript.

Reviewers' Comment: Additionally, I the ms would benefit from more general written description of trends in the results (or state that there are no trends). Currently the reader is left largely to pick their way through the massive, tiny-font tables.

Authors' Response: The reviewer is right. We have given more information (data ranges) to easier reading and interpretations of given results.

Reviewers' Comment: Title: should be "trace metal concentrations" Authors' Response: Corrected

Reviewers' Comment: P2, L2: what does it mean for an ecosystem to be "ecologically regulated"? Authors' Response: We meant that it hosts a particular and not random microbial (neston) community. We have removed these words and now the text reads: "Characterized by a high abundance of microorganisms (called neuston and ranging from bacteria to larger siphonophores (Wurl et al., 2017)), the SML constitutes a particular marine ecosystem."

Reviewers' Comment: P3, L15: should be "underlying" Authors' Response: Done

Reviewers' Comment: P4, L15: are the fine or the coarse particles <40 um? Authors' Response: We have modified the sentence to avoid confusion. The text now reads: "This inlet was developed to sample particles with an aerodynamic diameter inferior to 40 $\mu$m (Rajot et al., 2008)."

Reviewers' Comment: P4, L19: year for Rupprecht & Patashnick? Authors' Response: We have clarified the Rupprecht & Patashnick is a company.

Reviewers' Comment: P4, L24: what is meant by "the cruise smoking"? Authors' Response: We have modified the sentence. It now reads: "A sampling strategy was used to avoid contamination by the ship's fumes."

Reviewers' Comment: P4, L18: how was the sample collected off the glass plate? With a water or acid rinse? If so, how was dilution of original sample estimated? Authors' Response: The sample is collected from both sides of the glass plate directly into sampling bottles by using a Teflon wiper located in a PVC system (see attached pictures R2). Reviewers' Comment: P5, L22: double citation (also check for other instances of this) Authors' Response: We have removed all double citations along the manuscript.

Reviewers' Comment: P6, L4: disolved samples were not irradiated? Was a particulate CRM analyzed? What was the digestion approach for the particulate/total phase? How long were samples acidified for? Authors' Response: Dissolved fraction samples were not irradiated. We are aware that organic ligand and DOM are also present in the dissolved fraction. However, unlike other analytical methods where UV digestion is a critical step (e.g. polarography, or method using particulate chelating resin), in the method used here (liquid-liquid extraction using APDC-DDDC), the UV radiation is not considered a critical step. In the case of the total fraction we decided to include the UV step to guarantee a full cell breakdown and the complete digestion of samples. As

far as these coauthors know CRM for particulate metals in seawater are not available. For the total phase UV digestion we used an UV system consisting in one UV (80 W) mercury lamp that irradiated the samples (contained in quartz bottles) during 30 min. All samples (dissolved and total) were acidified and stored for at least 1 month prior to analysis. It has been now specified in the manuscript. The text now reads: "All samples were acidified on board to pH< 2 with Ultrapure-grade HCl in a class-100 HEPA laminar flow hood. The metals, (i.e. Cd, Co, Cu, Fe, Ni, Mo, V, Zn and Pb) were stored for at least 1 month prior to analysis."

Reviewers' Comment: P7, L4: I assume this should be 0.5 um Authors' Response: Yes, corrected.

Reviewers' Comment: P8, L28: here and later the text to refers to "previous studies" but only one study is cited Authors' Response: The reviewer is right. It has been corrected in the manuscript.

Reviewers' Comment: P9, L3: is it standard deviation or error presented here and in other parentheses? Authors' Response: It is standard deviation. It has been specified along the manuscript.

Reviewers' Comment: P10, L18: in the discussion of residence times the ms refers to particulate metals, but the methods only describes collecting a dissolved (filtered) and a total (unfiltered) fraction. So, how can the behavior of the particulate fraction be isolated and determined? Please explain the assumptions made to do this, so they can be evaluated. Authors' Response: We assumed that the metal in the total fraction is strongly influenced by lithogenic/aerosol material. We were aware that Total fraction also include metals from other pools (e.g. microbial and organic material) and perhaps, assuming this, we were overestimating the residence time. However, our residence time were of the same order that those calculated by Ebling and Landing (2017) using reactive and refractory particulate concentrations, and therefore making of our assumption feasible. We have specified this assumption in the manuscript, that

now reads: "For the calculations of the residence times throughout our different stations we used simultaneous empirical measurements of total metal concentrations in the SML (assuming that the metal in the total fraction is highly influenced by material from aerosol dust) and metal aerosol fluxes"

Reviewers' Comment: P11, L4: what does "dynamic" mean here? Authors' Response: We meant physical properties. We have modified the sentences that now reads: "….other variables (linked to physical processes, photochemistry or biological activity) probably affected the residence time of this and the other metals in the SML."

Reviewers' Comment: P11, L7: this is an example of the selective explanation of elemental behaviors. Is the influence of wind on just Co, or Co and Ni? Additionally, why would wind effect only one or two of the metals? Would wind not have the same physical transport or diffusion and mixing effects on all metals? Please provide some additional discussion of this very selective effect. Authors' Response: We agree with the reviewer. In deed the reviewer is right in his/her comment. We realized that we made a mistake in the residence time calculation because we used the aerosol metal flux of the first station (station 1) to estimate the residence time of all stations. We have revised and corrected all calculations and now residence time is calculated using the aerosol flux of each station (calculated using the aerosols concentration (table 1) and assuming a deposition velocity of 0.1cm/s). Recalculated residence time are of the same order of magnitude than before, however we changed some part of the discussion. In the previous version residence time of Co was very well correlated with wind speed, which opened the question on the lack of effect of wind speed on the other metals. Now the relative low wind speed during our campaign ($9 \pm 4.99$ knots) did not affect the residence time of metals in the SML. It has been indicated in this section.

Reviewers' Comment: P11, L11: it should be "nucleic acid" Authors' Response: Corrected

Reviewers' Comment: P11, L13: the methods for characterizing all of these separate

biological groups needs to be provided and justified: why were these groups characterized? Authors' Response: We have focused the discussion in the most representative microbial community in the SML (i.e. Bacteria; High nucleic acid-content bacteria: HNA; Low nucleic acid-content bacteria: LNA; pico-phytoplankton). We have removed the different phytoplankton groups from the text since discussion about their abundance is speculative with the existing data.

Reviewers' Comment: P11, L24: how is SML fraction different from T-SML? Authors' Response: We meant microbial abundance in the SML and reactive elements in the T-SML. We have modified the sentence to avoid confusion. Now the sentence reads: "Microbial abundance in the SML and reactive elements (i.e. Cu, Fe, and Zn) in the T-SML showed the same longitudinal gradients in this study,…."

Reviewers' Comment: P12, L2: bacterial assimilation would result in no change in T-SML, which includes both dissolved and particulate fractions. Authors' Response: The reviewer is right. This argument is valid only for the dissolved fraction, we have now removed the T-SML and the sentence now reads: "Bacteria could efficiently assimilate the fraction of Cu, Fe and Zn available, favouring a decrease in the D-SML fraction (Table 1-2)."

Reviewers' Comment: P12, L23: why would regeneration in the east only be active for Co, when many other bioactive metals are also remineralized actively? Authors' Response: The regeneration of biogenic particle is probably active for many other metals, however it only has been suggested in the Mediterranean Sea for Co.

Reviewers' Comment: P13: I think there is inadequate support for the conclusion that Ni is inhibiting growth in the surface waters, particularly given the lack of relationship with primary production and chl a Authors' Response: We agree and we have toned down the conclusion. Unfortunately, we don't have Chl-a and primary production measurements in the SML not even in the first meter of the sea surface to confirm our hypothesis (available Chl-a and primary production data are from 5 meter depth).

Reviewers' Comment: P14, L13: how can there be a 'major difficulty of mobility' (a strange term, I feel) for Fe when the residence times are only a few minutes (table 3)? Authors' Response: We agree. We have changed the word "mobility" by "diffusion".

Reviewers' Comment: Fig S2: please include the year in the date in the caption Authors' Response: Done

Reviewers' Comment: Fig S3: the figure does not match the caption Authors' Response: The reviewer is right. Figure S3 (now Figure S4) has been changed with rainfall rates from the radar European composite product that are geo-referenced allowing to plot the position of the FAST station. The figure cation has been changed consequently and now reads: "Accumulated rainfall during the night between June 3rd and 4th 2017 (00h00 – UTC) and position of R/V at the Fast Station. The rainfall rates are estimated from the radar European composite products provided by the Odyssey system." We really thank to the reviewer for all these comments and the many errors detected.

Please also note the supplement to this comment:
https://www.biogeosciences-discuss.net/bg-2019-290/bg-2019-290-AC2-supplement.pdf

———————————————————————

**Pictures R2**. SML system used during Pacetime campaign.

[Figure]

**Fig. 1.**

**Supplement:**

[revised manuscript text omitted]

---

## Author Response (AR2)

**Co-Editor-in-Chief Decision: Publish subject to minor revisions (review by editor)** (24 Feb 2020) by Christine Klaas
Comments to the Author (pdf): bg-2019-290-comments-to-author.pdf
Comments to the Author: Dear Dr. Tovar-Sánchez and co-authors.

Both reviewers have concluded that your manuscript is of high interest and suitable for publication in Biogeosciences. Based on the recommendations provided by the reviewers, I am happy to accept publication of a revised version of the manuscript that incorporates the reviewers' major comments (see also attached annotated manuscript); These are:

We really thank to this new referee for the effort made in the review process of this manuscript. We are sure that his/her constructive comments have helped to improve our manuscript. We have modified the text following her/his suggestions and incorporated further clarification where requested. The response and actions taken to accommodate the reviewers' comments are described in the following pages.

- "splitting the metal sources in between one type of big particles that bring only iron and ten times smaller particles that bring the rest of the metals is in my opinion unrealistic. The authors have done a better effort in trying to justify this approach but I still remain skeptic."

--> Provide residence-time values for Fe contemplating two scenarios (the new one would be to assume that iron could come from particles of the same origin and size) but this was ignored.

Response 1. In response to the previous reviewer, we believe that the use of big particle for Fe was well justified and unrebutted. However, we agree in the fact that the recalculated residence time of Fe using same particle size than rest of metals was ignored. The residence time calculation of Fe using the same deposition rate than the rest of metals (i.e. 0.1 cm/s instead 1cm/s) results in a residence time 10 times longer (i.e. average 0.6 h instead 0.06h), even so, the residence time of Fe remain being the shortest of all studied metals.

-" Ni toxicity. Here the authors brought again better arguments (Ni could generate toxic ROS and displace Fe from some enzymes) although this has not been tested in the ocean. However, I cannot accept that a simple correlation, where the concentration of metal spans by a factor of only 2, can be indicative of toxicity for bacteria. One thing is a simple suggestion but another matter is a discussion of more than one page that makes its way into the abstract. First, this type of line of reasoning assumes that correlation implies causation, second, toxic effects of trace metals are revealed when the concentration of the toxic element spans for at least an order of magnitude and here this factor is of less than 2 (Figure 2)."

Response 2. During the first round of this review, we agreed that simple correlation is not a valid proof to demonstrate Ni toxicity and for that reason we rewritten this finding as a "*hypothesis*", *toning down our statements and increasing the discussion to argument the hypothesis. We indicated that: "Although difficult to demonstrate with the existing data, we consider possible the toxicity of Ni by the following reasons: 1) the strongly and negative correlations between dissolved Ni (i.e. bioavailable) and microbial abundance in both, SML and SSW. This kind of correlation was not found with any other metal; 2) previous study demonstrated Ni toxicity in the same region although with concentrations 13 times higher that our measured background; and 3) we have included now the potential effect that intense UV radiation happening in the SML could have on the binding capacity of colloidal-Ni, and consequently on its bioavailability and/or potential toxicity. Even so, we are aware that this hypothesis remains speculative and we don't have enough information to demonstrate it."*
However, these arguments have been ignored by the reviewer. Since this argumentation remain unrebutted we consider this hypothesis valid although needs to be tested, which should result very useful for the oceanographic community.

- -> think a table should be added with more details than only the recovery on the chemical analyses with ICPMS. A table with the CASS values for example related to the defined concentration in the CASS sample.

Response 3. We have included the requested information in the text. The ms now reads: *"The accuracy of the pre-concentration method and analysis for trace metals was established using Seawater Reference Material for Trace Elements (CASS 6, NRC-CNRC) with the next concentrations (in µg/L) regarding the certified values (in brackets): Cd: 0.02 (0.022); Co: 0.07 (0.067); Cu: 0.44 (0.418); Fe: 1.58 (1.56); Mo: 8.1 (9.15); Ni: 0.41 (0.418); Pb:0.01 (0.011); V: 0.45 (0.49); Zn: 1.14 (1.127)."*

--> In the abstract the authors write about particulate metals, these were not determined. Unfiltered water was analysed, so call these values unfiltered concentrations. Particulate metals were not sampled and were as far as I understand also not calculated by subtracting dissolved from total concentrations (which I do not recommend, let that be clear).

Response 4. The reviewer is right, only dissolved fraction and unfiltered fractions were measured in the SML. It has been clarified along the text. However, in the abstract we refer to the calculated residence times of particulate metals derived from aerosols deposition, which was calculated using the Total (unfiltered fraction) of the SML.

In the conclusion it reads:
"While some metals entering the SML (e.g. Cd, Co, Ni and V) show efficient diffusive mixing from the SML to the SSW, more reactive metals such as Cu, Fe, Pb and Zn seem to exhibit a slower diffusion."

--> Diffusion and diffusive mixing was not discussed, residence times were. And these residence times show that the residence time of Cu is one of the largest whereas the one of Fe is the shortest, as is also stated in the abstract. Still Fe and Cu both seem to exhibit a slower diffusion. This has to be explained better or if not supported by the results changed into another conclusion

Response 5. The reviewer is right. Since we don't have enough data to confirm this conclusion we have changed/replaced it and now reads: *"Residence times of particulate metals derived from aerosols deposition were highly variable, with the largest residence times for Cu (5.8 ± 6.2 h) and the lowest for Fe (3.6 ± 6.0 min)"*

Quote
" 3.2.1. Trace metals in the SML
Dissolved concentrations of Co, Zn, Pb, Cu and Ni showed a decreasing trend from the SML to the SSW, with average concentrations (± SD) 10.4 ± 0.7, 9.3 ± 5.5, 4.2 ± 1.8, 3.1 ± 1.5, and 1.2 ± 0.1 times higher in the SML than in the SSW, respectively. The SML to SSW concentration ratio for V (1.2 ± 0.42) and Fe (1.3 ± 1.5) indicated only slight enrichment in the SML over the underlying water, while the ratio for Mo (1.0 ± 0.1) indicated no difference between layers (Table 1). Only Cd concentrations were consistently lower in the SML compared to the underlying water (0.8 ± 0.2 times lower). Such depletion of dissolved metals in the SML compared to the underlying water has been previously observed in areas without significant aerosol inputs (Ebling and Landing, 2015, 2017). Although not fully understood, some mechanisms such as the dominance of removal mechanisms versus diffusion, or the higher influence of underlying metal sources have been suggested previously to explain this metal depletion (Ebling and Landing, 2017; Hunter, 1980)."

--> Ni enrichment is lower (significantly?) than Fe but equal (significantly?) to V

Response 6. We are not sure if we understood the reviewer criticism. We did not aim to make a comparison between elements but between compartments for each element. In this sense we can confirm that the differences of one metal between compartments are or not significantly different.

Second: Such depletion: Only Cd has a lower ratio. So is it Cd the authors write about or is the enrichment factor so low compared to ??? that the word depletion is used. This needs explanation or a different reasoning.

Response 7. The reviewer is right. We referred to Cd and it has been clarified in the ms that now reads: "Such low ratio of dissolved metals (as it is the case of Cd) in the SML compared to the underlying water has been..…"

Third : Apart from my confusion about this, the actual word depletion in my book has a relation to not enough for biology, blooms stop. Perhaps the word depleted must be changed by low ratio, low concentration. (but the above first and second comments, remain)

--> see annotated manuscript

Response 8. To avoid confusion, we have changed two sentences where the word "depletion" was used. Now, the ms reads:
- "Such low ratio of dissolved metals (as it is the case of Cd) in the SML compared to..…"
- "….the dominance of removal processes over diffusion, or the higher influence of metal sources from underlaying water have been suggested previously to explain this difference of metal concentrations between compartments …"

Pae 11 lines 11 onwards
Quote: "During the cruise, Al and Fe atmospheric concentrations were correlated at all the stations and the ratio Fe/Al is typical of a crustal source (Fu et al., in prep.). It is known that the atmospheric iron deposition fluxes are associated to mineral dust particles even during the period when the Saharan dust inputs are very low (Desboeufs et al., 2018; Guieu et al., 2010). On the contrary, no correlation with Al is observed for the other metals, except during FAST1-3. Thus, we used a velocity of mineral dust deposition for Fe 1 cm.s- 1 and an average velocity of fine anthropogenic particles for the other metals, i.e. 0.1 cm.s-1 (Baker et al.,2010; Duce et al., 1991)."
It is all very well that Fe correlates with Al, but there is hardly transport from the south. And this Fe/Al ratio, is that not related to the mineral lattice? So not really Fe that will dissolve so easily in such a short residence time. Moreover, this factor 10 difference in velocity decreases the residence time with a factor 10. If the authors had used the same deposition for Fe, the residence time ± the error would overlap the residence time of Co and Zn. And then only Ni would have a residence time that is extremely short, which would fit the lack of reactivity of this element in contrast to an element as Fe. This might interfere with the role of Ni in the discussion though.

--> see previous comments and annotated manuscript

Response 9. We have clarified this point in previous comments and we hope have provided enough arguments to convince the reviewer that about our decision on this issue.

Lines 17-21
Quote: " Since, such a quick transfer of these metal particles to the underlying water (in the order minutes) is unlikely (mainly due to their affinities to organic ligands), and the dissolution is not immediately reflected in an increase in the concentration in the dissolved fraction (i.e. D-SML), other variables (linked to physical processes, photochemistry or biological activity) probably affected the residence time of this and the other metals in the SML."
Gerringa et al, 2017 ((mar. chem. 194, 100-113) concluded that ligands are saturated in the surface Mediterranean so that might form one explanation that a quick transfer as particles to the underlying water is possible.One of the co-authors of this manuscript, Wagener (in Wagener et al. 2010), might have some opinions here, that could explain this quick transfer also very well!

Response 10. The argument of the reviewer could be feasible. Nevertheless, this argument would add more speculation to the work since our dataset is not comparable with that obtained by Gerringa et al, where surface data were obtained at 9 m depth and our values are from the SML and 1 m depth.

Quote: "On average, while the highest residence times obtained for Cu and Pb are in agreement with their strong affinity to particles and therefore with a high probability of retention in the SML, other reactive elements such as Fe presented the shortest residence times. Since, such a quick transfer of these metal particles to the underlying water (in the order minutes) is unlikely (mainly due to their affinities to organic ligands), and the dissolution is not immediately reflected in an increase in the concentration in the dissolved fraction (i.e. D-SML), other variables (linked to physical processes,

photochemistry or biological activity) probably affected the
residence time of this and the other metals in the SML" .

I cannot follow the reasoning: the long residence times of Cu and Pb are explained by a strong affinity
to particles. However, one of the possible explanations for a short residence time is also binding by
particles (a quick transfer to particles). Although according to the authorsthis is not the case for Fe, it
still served as a possible explanation for a short residence.

Response 11. We agree with the reviewer, and we have rewritten the sentence that now reads: "*Since
such a quick transfer of these metal particles to the underlying water (in the order minutes) is unlikely
(mainly due to the high content of organic matter in the SML), and the dissolution is not immediately
reflected in an increase in the concentration in the dissolved fraction*".

Page 14:
Quote: "Since surface salinity showed the same eastward increase and was closely correlated with
those metals (rs ranged from 0.51 p<0.05 for Mo to 0.97 p<0.01 for Ni; Table 3), the exchange with
the surface Atlantic Ocean waters seems to be the main cause of this gradient of concentrations in our
study, although higher aerosol inputs in the western MS could also contribute to this gradient."

How can more aerosol in the west decrease dissolved concentrations? I think dissolution would
increase the conc, if you mean ballasting and adsorption processes are connected with the decrease in
dissolved concentrations than this must be better explained.

Response 12. The reviewer is right. We meant eastern instead western which was in agreement with
the statement made before (i.e. "*This trend is consistent with previous studies where the increasing
eastward trend in concentration along the southern coast of the MS has been suggested to result from
several factors such as:  more intense Saharan deposition on the eastern MS (Guieu et al., 2002)*").
To avoid to be repetitive we have deleted "..,although higher aerosol inputs in the eastern MS could
also contribute to this gradient"

Minor comments:
Abstract, Line 33: Ni concentration: do you mean total or dissolved ? Since both are discussed in the
lines just above.

Response 13. We meant both Dissolved and total (unfiltered). It has been specified in the abstract
that now reads: "*Our results show a strong negative correlation between the dissolved and total Ni
concentration and heterotrophic bacterial abundance in the SML and SSW,…*".

Page 2: lines 17-19
Copied from the Conclusion in Sarthou and Jeandel 2001: "The very low surface concentrations may
limit primary production or, at least control phytoplankton species composition. Such an iron depletion
was not expected in the Mediterranean Sea, characterised by important continental inputs."

For me "may limit" is not equal to saying that " Fe has been considered an important factor controlling
phytoplankton growth (Sarthou and Jeandel, 2001)."

So tone down here please.

Response 14. We have rewritten the sentence, now reads: "Although present in a higher concentration
than in other oceans, Fe may control phytoplankton growth and composition (Sarthou and Jeandel,
2001)."

Page 9:
Line 25, were the metals in the FAST station comparable then with Tovar-Sanchez et al., 2014?

Response 15. Yes, for most of them (i.e. Co, Cu, Fe, Mo, V).

Line 17 however, the strong dependence of redox seawater chemistry and complexation of elements such as Cu and Fe on solar radiation is well known (Croot and Heller, 2012; Moffett and Zika,1988).
I would not say that this is really well known, some studies have been done, true but in my opinion nothing on the combination of complexation and redox chemistry is well known.
Please change "well known" in "has been studied" or some knowledge is present or another rewording.

Response 16. We agree, we have changed the "well known" by "have been studied".

Page 12, lines12-14
Our results indicate that Fast 1-4 stations were affected by the dusty rain events, which increased the concentration of some metals in the T-SML and consequently the residence time (Table 4). However, the reasons for the increase at stations 3-4 are not evident.

For me this is unclear, 1-4 means 1,2,3,4. So if 1-4 can be explained then why 3 and 4 not?
--> see also annotated manuscript.

Response 17. The text refers to Fast 1-4 (Fast1, Fast1, Fast 3 and Fast4) and St3 and St4. We have clarified this in the ms, and now reads: "*However, the reasons for the increase at stations St 3 and St 4 are not evident*".

Regarding **annotated comments in the manuscript**, we have corrected all grammatical suggestions and clarified the next specific comments:

Review´s comments – Pag 2 -Line 11. "What do the authors mean by consistent?"
Response to reviewer´s comment: By consistent we meant that are usually present.

Review´s comments – Pag 2 -Line 21. "proxies for atmospheric deposition in Posidonia?"
Response to reviewer´s comment: aerosols in Posidonia tissue.

Review´s comments – Pag 2 -Line 23. "Meaning of this sentence is vague. Is it really necessary?"
Response to reviewer´s comment: We agree. It has been deleted.

Review´s comments – Pag 2 -Line 28. "not necessary?"
Response to reviewer´s comment: We agree. It has been deleted.

Review´s comments – Pag 3 -Lines 14-16. "Is this for the whole MS? Seems little. How relevant is that compared to the amount of Fe in the surface mixed layer?"
Response to reviewer´s comment:Yes, it was calculated for the whole Mediterranean Sea. 2 tons of total (unfiltered) Fe represents only 0.2 of the TSSW pools over the top meter of the surface sea. This amount would be much lower if we compare it with the surface mixed layer.

Review´s comments – Pag 3 -Line 18. "impact on the biogeochemistry of
Response to reviewer´s comment: it has been changed

Review´s comments – Pag 3 -Line 23. "range"
Response to reviewer´s comment: it has been changed

Review´s comments – Pag 5.
Response to reviewer´s comment: All suggested changes have been corrected in page 5.

Review´s comments – Pag 6 -Lines 7-8. "at what depth where the samples collected? Same as for PP? (see p. 7, line 21)"
Response to reviewer´s comment: SSW were collected at 1 m depth (as indicated at the beginning of the first paragraph of the section 2.1.), however, to avoid confusion we included the depth in this sentence ("SSW (~1 m depth) were collected using acid-washed Teflon tubing connected to a peristaltic pump and directly filtered on the same cartridge to collect the dissolved fraction (D-SSW)").

Samples for Primary Production (and Chl-a) were collected at 5m depth, which is indicated in section 2.3.2 and discussed in the last paragraph of section 3.2.4.

Review´s comments – Pag 8 -Line 5. "references?"
Response to reviewer´s comment: This data was obtained from the cruise and are pending of publication.

Review´s comments – Pag 9 -Line 5. "I could not find any analysis including aerosol data in Table 3."
Response to reviewer´s comment: The correlation was not showed but can be done and checked with data from Table 1.

[Figure]

Review´s comments – Pag 9. Line 9: "What is the unit for the color scale in Fig. S4?"
Response to reviewer´s comment: The units are mm/period, it has been included in the legend of the figure.

Review´s comments – Pag 9. Lines 9-12: "I am not sure I follow the logic. How is the composition related to the duration of rain events?"
Response to reviewer´s comment: The reviewer is right, the sentence was not clear. This sentence is to justify, not the duration but the type of wet deposition, as the sampled rain is a dusty rain, we suppose that all the rains occurred during the fast action are dusty rains. Moreover, we used the argument of high metals concentration in rain to justify the high concentration in the surface water of Fast stations. The text now reads: "As the rain composition collected was typical of wet dust deposition with high particulate concentrations of Al, Fe and Ca (Fu et al., in preparation), we suppose the rain-out of dust in the atmospheric column around this station occurred between the 3rd to the 5th of June and hence dusty rain events impacted the concentration of surface waters during Fast stations."

Review´s comments – Pag 9 Line 20: "Why is there no analysis of trend between T-SLM and D-SLM and T-SLM and D-SSW (Table 3)?"
Response to reviewer´s comment: T-SML vs DSSW analysis is included in Table3, however comparison between T-SML and D-SLM was not included since maximum correlation were only found for D-SML Cd and T-SML Cd (i.e. 0.789 p<0.01), however the correlations of D-SML and T-SML versus biology parameters in SML and SSW were done and presented in Table 5.

Review´s comments – Pag 10 Line 12: "Meaning unclear. Do you mean metal sources in the underlying water layer?".
Response to reviewer´s comment: We meant metal sources from underlaying water. It has been rewritten, and the text now reads: "or the higher influence of metal sources from underlaying water have been suggested previously…"

Review´s comments – Pag 10 Lines 2-4: "See reviewers comments. Since there is no evidence here that Al and Fe fluxes come from other sources; reviewer 1 has a fair point in asking for the presentation and discussion of residence times using similar sinking velocities as for other metals."

Response to reviewer´s comment: This have been responded above (Response 1).

Review´s comments – Pag 12 Line 14: "From Fig. S1 and S2 it seems that the source regions for st.3 and 4 are the same as for the FAST location?"
Response to reviewer´s comment: Perhaps the source regions could be similar or close however only FAST stations were affected by run event which help to explain the high metal concentration measured in the SML.

Review´s comments – Pag 13 Lines 4-7: "From Table 2, I do not see evidence of increased bacterial abundances due to the dust input: st. 3 and 4 also have higher bacterial abundance."
Response to reviewer´s comment: The reviewer is right, we have removed the sentence. Also, we have include more references in this section to support our findings.

Review´s comments – Pag 13 Lines 13-16: "Contradicts argument proposed p. 14 lines 5-11."
Response to reviewer´s comment: We don't find contradictory this argument. Here we found that metals in the T-SML (i.e. unfiltered samples, and therefore including bacterioneuston) decreased eastward and was positive correlated with bacterioneuston abundance. In page 14 lines 5-11, we refer to metals concentrations in a different compartment and fraction (i.e. dissolved metals in the subsurface water) and therefore affected by different processes.

Review´s comments – Pag 13-14 Lines 26-27, 1-2 "See reviewers comments. Bacteria and Ni are correlated with salinity, hence, this could be an effect of macronutrients (see p. 13, line 10)."
Response to reviewer´s comment: We already explained this point to the previous reviewer.
The difference found with Ni regarding other metals is that the negative correlation with microbial abundance occurs in both compartments, SML and SSW, and in both fractions, dissolved and total. In the case of total fraction, that includes bacterioplankton metal composition, an uptake of Ni as nutrient should be reflected in an increase of abundance and also in an increase of Ni concentration in the total fraction. Also, the concentration of Ni in the SML is too high to act as nutrient for bacterioplankton.

Review´s comments – Pag 15: "See reviewers comments. This discussion is long as it is based on speculation (Both Ni and bacteria are correlated with salinity and could be the result of other environmental gradients (see previous comments on this and p. 13)."
Response to reviewer´s comment: We have clarified this point above (please, see Response 2).

Review´s comments – Pag 25: "these units do not correspond to those in the Table."
Response to reviewer´s comment: The reviewer is right. It has been corrected and the Table legend norw reads: "Arcillary parameters and metals in water and aerosols measured at all stations. Metals in ng.m$^{-3}$ for aerosols and nM or pM for water compartments"

---

## Editor Decision (ED3)

[revised manuscript text omitted]
 | -,602* | -,502* | -,534* | -0.113 | 0.365 | -,726** | -,591* | -,664** | -0.287 | 0.421 | 0.024 | -0.251 |

---

## Editor Decision (ED4)

[revised manuscript text omitted]
 | -,602* | -,502* | -,534* | -0.113 | 0.365 | -,726** | -,591* | -,664** | -0.287 | 0.421 | 0.024 | -0.251 |